# SMC5/6 is required for replication fork stability and faithful chromosome segregation during neurogenesis

Alisa Atkins[†], Michelle J Xu[†], Maggie Li, Nathaniel P Rogers, Marina V Pryzhkova*, Philip W Jordan*

Biochemistry and Molecular Biology Department, Johns Hopkins University Bloomberg School of Public Health, Baltimore, United States

**Abstract** Mutations of SMC5/6 components cause developmental defects, including primary microcephaly. To model neurodevelopmental defects, we engineered a mouse wherein *Smc5* is conditionally knocked out (cKO) in the developing neocortex. *Smc5* cKO mice exhibited neurodevelopmental defects due to neural progenitor cell (NPC) apoptosis, which led to reduction in cortical layer neurons. *Smc5* cKO NPCs formed DNA bridges during mitosis and underwent chromosome missegregation. SMC5/6 depletion triggers a CHEK2-p53 DNA damage response, as concomitant deletion of the *Trp53* tumor suppressor or *Chek2* DNA damage checkpoint kinase rescued *Smc5* cKO neurodevelopmental defects. Further assessment using *Smc5* cKO and auxin-inducible degron systems demonstrated that absence of SMC5/6 leads to DNA replication stress at late-replicating regions such as pericentromeric heterochromatin. In summary, SMC5/6 is important for completion of DNA replication prior to entering mitosis, which ensures accurate chromosome segregation. Thus, SMC5/6 functions are critical in highly proliferative stem cells during organism development.

**\*For correspondence:**
mpryzhk1@jhu.edu (MVP);
pjordan8@jhu.edu (PWJ)

[†]These authors contributed equally to this work

**Competing interests:** The authors declare that no competing interests exist.

## Introduction

Preservation of genomic integrity is crucial for normal organism development and homeostasis. Structural maintenance of chromosomes (SMC) complexes (cohesin, condensin, and SMC5/6) function as guardians of chromosome architecture and genomic stability (*Hagstrom and Meyer, 2003*; *Kschonsak and Haering, 2015*; *Lehmann, 2005*; *Remeseiro and Losada, 2013*; *Uhlmann, 2016*). During DNA replication, newly synthesized sister chromatid DNA is held together by cohesin (*Morales and Losada, 2018*). Removal of cohesin is essential for mediating chromosome segregation (*Batty and Gerlich, 2019*). Condensin complexes are required for ordered compaction of chromatin to facilitate the formation of condensed chromosomes prior to chromosome segregation (*Hirano, 2016*; *Kagami and Yoshida, 2016*). SMC5/6 is the least characterized of the three SMC classes, but studies in yeast and mammalian cell lines have demonstrated that the complex is important for response to DNA replication stall/block, mediating DNA repair, and ensuring accurate chromosome segregation (*Aragón, 2018*; *Verver et al., 2016*).

The SMC5/6 complex is composed of a heterodimer of SMC5 and SMC6, which interact at their central hinge domains. SMC5 and SMC6 harbor long coiled coil domains emanating from both sides of the hinge domain that fold back on each other to form an ATPase at the juxtaposed C and N termini (*Hassler et al., 2018*). The ATPase head/tail domains of SMC5 and SMC6 are bridged by NSMCE4A (non-SMC element 4A), a kleisin protein. NSMCE4A interacts with NSMCE1, an E3 ubiquitin ligase, and NSMCE3, a MAGE (melanoma-associated antigen gene) domain-containing protein (*Palecek et al., 2006*; *Pebernard et al., 2008a*). Additionally, the SMC5/6 complex comprises an E3 SUMO ligase, NSMCE2, which interacts with a region of the coiled coil domain of SMC5

(*Andrews et al., 2005*; *Zhao and Blobel, 2005*). Two additional proteins, SLF1 and SLF2 (SMC5/6 localization factors), form a dimer that primarily interacts with the SMC5/6 complex during DNA repair in response to DNA lesions that block DNA replication (*Räschle et al., 2015*). SLF2 binds to the arms of SMC5 and SMC6 and may lock the arms together to immobilize SMC5/6 at sites of DNA damage (*Adamus et al., 2020*; *Räschle et al., 2015*).

Mutations within genes encoding SMC complex components are associated with human developmental defects. Collectively known as cohesinopathies, mutations in cohesin components lead to an array of abnormalities, including growth retardation and cognitive impairment (*Boyle et al., 2015*; *Piché et al., 2019*). Mutations in condensin components lead to primary microcephaly due to chromosome decatenation failure during mitosis (*Martin et al., 2016*; *Nishide and Hirano, 2014*). In regard to SMC5/6, mutations in *NSMCE3* result in immunodeficiency and lung disease, where patient-derived cells exhibit hallmarks of chromosome instability and replication stress (*van der Crabben et al., 2016*). Moreover, mutation of *NSMCE2* causes primordial dwarfism and primary congenital microcephaly (*Payne et al., 2014*). Assessment of patient cells harboring the *NSMCE2* mutation revealed chromosome instability and increased sensitivity to DNA replication stress (*Payne et al., 2014*). Thus, the microcephaly is likely the consequence of neural progenitor cell (NPC) depletion due to compromised genomic integrity. Furthermore, genetic variations in other components of the SMC5/6 complex, including *SMC5*, are potentially associated with congenital defects including heart and neurodevelopmental anomalies (*Homsy et al., 2015*; *Jin et al., 2017*; *Landrum et al., 2018*).

Despite the relevance of SMC5/6 mutations to human health, in vivo studies in mammalian models remain limited. One study found that conditional knockout (cKO) of *Nsmce2* in mice during adulthood causes premature aging and susceptibility to cancer (*Jacome et al., 2015*). Analysis of cell cultures from mice with *Nsmce2* mutation revealed increased formation of micronuclei and sister chromatid exchange events (*Jacome et al., 2015*). Two other studies have focused on using *Smc5* cKO to address the sexually dimorphic roles of SMC5/6 during gametogenesis, wherein SMC5/6 is largely dispensable for spermatogenesis but is essential for mediating chromosome segregation during oogenesis (*Hwang et al., 2018*; *Hwang et al., 2017*). Null mutations of SMC5/6 components in mice result in a failure to reach blastocyst stage (*Hwang et al., 2017*; *Jacome et al., 2015*; *Ju et al., 2013*). Thus, the roles of SMC5/6 during later stages of embryonic development have not been assessed. Because of the link between SMC5/6 perturbation and neurodevelopmental disorders in humans (*Homsy et al., 2015*; *Jin et al., 2017*; *Landrum et al., 2018*; *Payne et al., 2014*), we modeled the consequences of SMC5/6 depletion by conditionally mutating *Smc5* in the developing neocortex of mice.

Development of the cerebral cortex is a remarkably complex process that relies on the capacity of NPCs to undergo a series of coordinated cell division, migration, and differentiation steps. NPCs reside in the ventricular zone (VZ) of the cerebral cortex and undergo symmetric and asymmetric divisions to self-renew and produce intermediate progenitors (IPs) or neurons. IPs are mainly located in the subventricular zone (SVZ) (*Kowalczyk et al., 2009*; *Paridaen and Huttner, 2014*). During embryonic development, apical NPCs and IPs divide to produce neurons, which then migrate in the process of cortical lamination. This migration occurs in an inside-out manner, in which early-born neurons give rise to deep cortical layers (V and VI) and late-born neurons form superficial layers (II–IV) (*Molyneaux et al., 2007*; *Paridaen and Huttner, 2014*; *Shibata et al., 2015*). Disruption of NPC genomic integrity during embryonic development causes increased NPC apoptosis, reduced neuron production, and neuron mislocalization, ultimately resulting in decreased cortex size or microcephaly (*McKinnon, 2013*). The rapid proliferative activity of NPCs imparts a high degree of endogenous replication stress and DNA damage, which can lead to the formation of excess single-stranded DNA (ssDNA) and double-strand breaks (DSBs) (*Harley et al., 2016*; *Lee et al., 2012*; *McKinnon, 2017*; *O'Driscoll, 2017*; *Reynolds et al., 2017*). The failure to complete DNA repair and resolve replication intermediates may contribute to chromosome segregation errors and p53-mediated apoptosis (*Mankouri et al., 2013*; *Rodrigue et al., 2013*).

The DNA damage response (DDR) is a first line of defense against insults to genome integrity in the nervous system. ATM and ATR kinases play independent and essential roles in DDR, and their loss can lead to neurodevelopmental disorders and neurodegeneration (*Enriquez-Rios et al., 2017*; *Madabhushi et al., 2014*). ATM can be activated by DSBs in both NPCs and immature neurons, while ATR is responsible for G2/M checkpoint induced by RPA-bound ssDNA during replication

stress in proliferating NPCs (*Enriquez-Rios et al., 2017*). Both kinases can initiate either DNA repair or cell apoptosis. DNA DSB repair by homology-directed repair (HDR) is prevalent in mitotically active NPCs, and non-homologous end joining becomes the pathway for DNA repair in postmitotic neurons (*Enriquez-Rios et al., 2017*; *Madabhushi et al., 2014*). ATR and ATM act through their respective downstream targets CHEK1 and CHEK2 to activate p53-mediated cell cycle arrest or apoptosis if DNA damage remains unrepaired (*Sengupta and Harris, 2005*; *Shaltiel et al., 2015*).

In this study, we show that depletion of SMC5 at early stages of mouse brain development resulted in reduced cerebral cortex size due to mitotic abnormalities in NPCs, perturbed differentiation, and increased apoptosis, which collectively affected cortical layer formation. We observed that SMC5 depletion leads to genome instability and DDR activation. We found that the cortex size of *Smc5* cKO mice was restored upon knockout of *Trp53* or *Chek2* genes. Subsequently, using an auxin-inducible degron (AID) system to deplete SMC5 in mouse embryonic stem cells (mESCs) we mimicked cell proliferation and cell death phenotypes observed in vivo, which were also rescued by inhibition of p53 or CHEK2. Using the AID system, we determined that SMC5/6 is important for DNA replication fork stability and propose a role for SMC5/6 in the processing of under-replicated DNA intermediates to ensure accurate chromosome segregation.

## Results

### *Smc5* cKO causes neurodevelopmental and sensorimotor defects due to aberrant chromosome segregation and apoptosis of NPCs

To model the consequences of SMC5/6 depletion during neurodevelopment, we used mice harboring a *Smc5* cKO allele (*Figure 1A*). Exon 4 of the *Smc5* cKO allele was flanked by *loxP* Cre recombinase target sequences. Depletion of SMC5 via *Smc5* cKO has been shown to destabilize the entire SMC5/6 complex (*Gaddipati et al., 2019*; *Hwang et al., 2018*; *Hwang et al., 2017*; *Pryzhkova and Jordan, 2016*). We employed two different Cre recombinase transgenes driven by tissue-specific promoters *Nestin-Cre* and *Emx1-Cre*. *Nestin-Cre* is expressed throughout the central nervous system (*Dubois et al., 2006*). *Emx1-Cre* expression is restricted to the mouse cerebral cortex (*Gorski et al., 2002*). *Nestin-Cre* and *Emx1-Cre* recombinases demonstrate robust expression in the mouse cortex at E12.5, and the use of either one to mediate *Smc5* cKO was expected to result in a similar phenotype during cortex development (*Insolera et al., 2014*).

First, we evaluated the efficiency and associated phenotype of *Nestin-Cre*-mediated *Smc5* cKO and depletion of SMC5 protein in E13.5 and E16.5 embryos (*Figure 1A–G* and *Figure 1—figure supplement 1*). Although we observed conditional mutation of *Smc5* in E13.5 brains, the protein levels of SMC5 and SMC6 were only partially diminished (*Figure 1—figure supplement 1A,B*). In contrast, the deletion in *Smc5* was accompanied by a substantial reduction in SMC5 and SMC6 protein levels in E16.5 embryos (*Figure 1A–C*). Since Nestin is expressed in progenitor cells of tissues other than neural, some degree of DNA excision was also observed in kidneys and muscles in our samples (*Figure 1B* and *Figure 1—figure supplement 1A*; *Bernal and Arranz, 2018*; *Sakairi et al., 2007*). SMC5 depletion in the developing mouse cortex resulted in a significant increase in cleaved caspase 3 (CASP3)-positive cells in *Smc5* cKO compared to controls in both E13.5 and E16.5 cortices (*Figure 1D,E* and *Figure 1—figure supplement 1C,D*). The increase in the number of cells undergoing apoptotic DNA fragmentation was most pronounced in E16.5 cortices (*Figure 1F,G* and *Figure 1—figure supplement 1E,F*).

We have previously shown that the depletion of SMC5 in mESCs causes chromosome segregation defects during mitosis (*Hwang et al., 2017*; *Pryzhkova and Jordan, 2016*). Thus, we analyzed VZ and SVZ mitotic progenitors in cortical sections of E13.5 and E16.5 embryos in detail (*Figure 1H,I* and *Figure 1—figure supplement 1G–I*). The percentage of NPCs undergoing abnormal mitosis with characteristic DNA bridges and lagging chromosomes at anaphase was elevated in *Smc5* cKO compared to control (*Figure 1H,I* and *Figure 1—figure supplement 1G–I*). We observed an increase in abnormal anaphase cells when comparing *Smc5* cKO cortices from E13.5 to E16.5 (9% versus 38%, respectively; *Figure 1I* and *Figure 1—figure supplement 1H*).

NPCs residing at the VZ apical surface undergo two modes of cell division, symmetric for self-renewal and asymmetric for differentiation (*Lancaster and Knoblich, 2012*; *Shitamukai and Matsuzaki, 2012*). It has been shown that NPC mitotic spindle orientation determines the positioning of

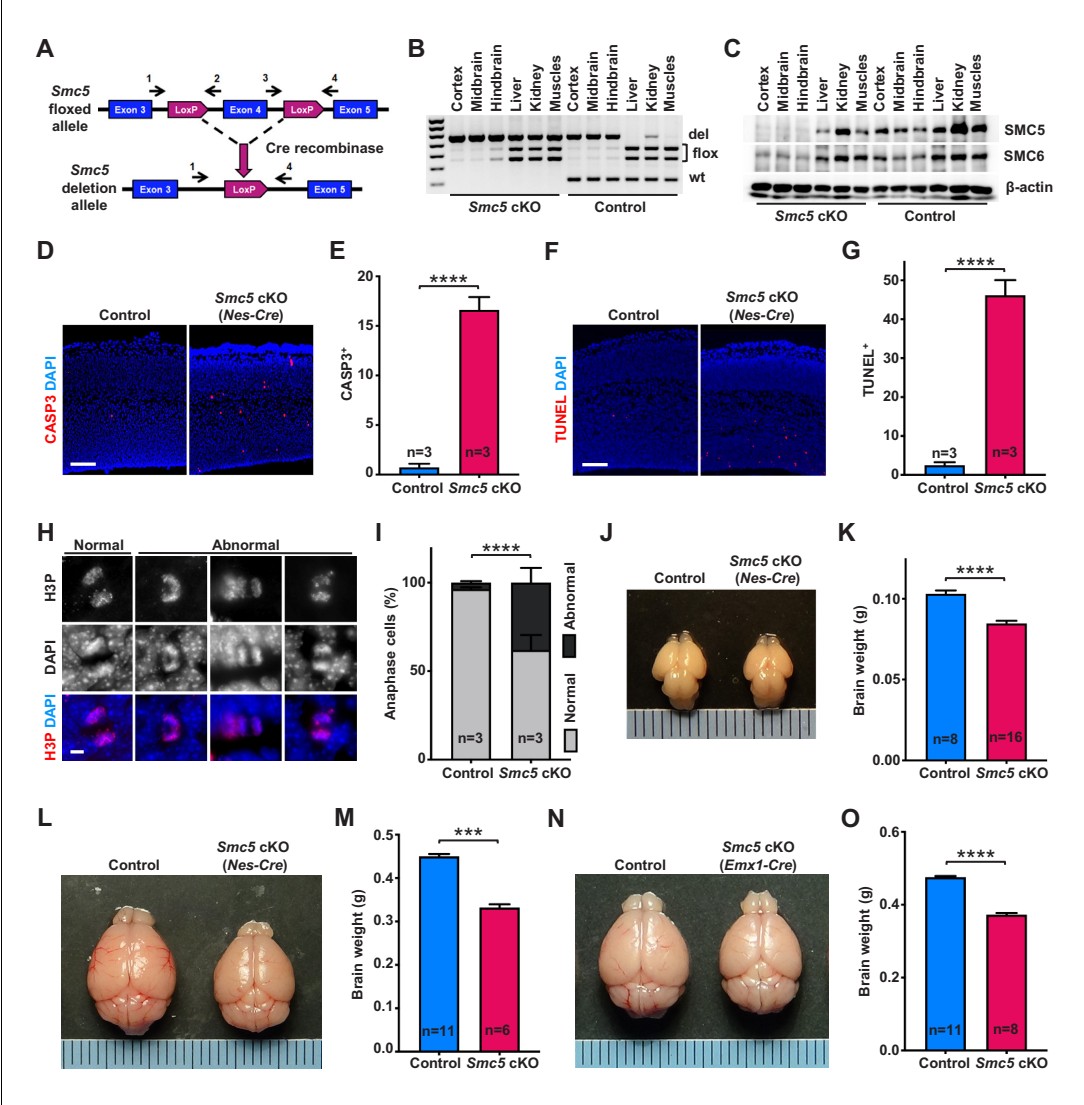

**Figure 1.** *Smc5* conditional knockout (cKO) causes disruption of genomic integrity and apoptosis of neural progenitor cells (NPCs). (**A**) The scheme of mouse *Smc5* floxed allele and Cre recombinase-mediated exon four excision. Arrows with numbers above represent genotyping primers (see Materials and methods). (**B**) PCR genotyping of control and *Smc5* cKO (*Nestin-Cre*) E16.5 tissues. Abbreviations: wt, wild type; flox, floxed allele; del, deletion. (**C**) Western blot analysis of SMC5 and SMC6 protein expression in control and *Smc5* cKO (*Nestin-Cre*) E16.5 tissues. β-actin was used as a loading control (n = 3). (**D**) Representative images of cleaved caspase 3 (CASP3) (red) staining in control and *Smc5* cKO (*Nestin-Cre*) E16.5 sagittal brain sections; DAPI (blue). Column width: 500 µm, scale bar: 100 µm. (**E**) Quantification of CASP3+ cells within 300 µm columns in brain sections related to (**D**). Data represent mean ± S.E.M. (control animals n = 3, *Smc5* cKO animals n = 3; see ***Supplementary file 3*** for details). Unpaired two-tailed Mann–Whitney test, ****p<0.0001. (**F**) Representative images of TUNEL+ nuclei (red) in control and *Smc5* cKO (*Nestin-Cre*) E16.5 coronal brain sections; DAPI (blue). Column width: 500 µm, scale bar: 100 µm. (**G**) Quantification of the number of TUNEL+ nuclei within 800 µm columns in brain sections related to (**F**). Data represent mean ± S.E.M. (control animals: n = 3, *Smc5* cKO animals: n = 3; see ***Supplementary file 3*** for details). Unpaired two-tailed Mann–Whitney test, ****p<0.0001. (**H**) Representative images of normal and abnormal mitotic E16.5 NPCs in anaphase, stained with an antibody against phospho-histone H3Ser10 (H3P) (red); DAPI (blue). Scale bar: 5 µm. (**I**) Quantification of percentage of normal and abnormal mitotic E16.5 NPCs in control and *Smc5* cKO (*Nestin-Cre*) brain sections. Data represent weighted mean ± weighted S.D. (control anaphase cells n = 85 from three animals, *Smc5* cKO anaphase cells n = 79 from three animals, see ***Supplementary file 3*** for details). Pearson's chi-squared test with Yates' continuity correction, ****p<0.0001. (**J**) Images of control and *Smc5* cKO (*Nestin-Cre*) P1 brains. Metric ruler is provided for scale. (**K**) Quantification of control (n = 8) and *Smc5* cKO (n = 16) (*Nestin-Cre*) P0/1 brain weight. Data represent mean ± S.E.M. Unpaired two-tailed Mann–Whitney test, ****p<0.0001. (**L**) Images of control and *Smc5* cKO (*Nestin-Cre*) P56 brains. Metric ruler is provided for scale. (**M**) Quantification of control (n = 11) and *Smc5* cKO (n = 6) (*Nestin-Cre*) P56-59 brain weight. Data represent mean ± S.E.M. Unpaired two-tailed Mann–Whitney test, ***p=0.0002. (**N**) Images of control and *Smc5* cKO (*Emx1-Cre*) P56 brains. Metric ruler is provided for scale. (**O**) Quantification of control (n = 11) and *Smc5* cKO (n = 8) (*Emx1-Cre*) P55-56 brain weight. Data represent mean ± S.E.M. Unpaired two-tailed Mann–Whitney test, ****p<0.0001.

The online version of this article includes the following figure supplement(s) for figure 1:

*Figure 1 continued on next page*

*Figure 1 continued*

**Figure supplement 1.** Evaluation of SMC5 depletion phenotype in embryonic cortices.
**Figure supplement 2.** *Smc5* conditional knockout (cKO) results in reduced cortex and brain size.
**Figure supplement 3.** Reduced brain size in *Smc5* conditional knockout (cKO; *Nestin-Cre*) mice affects sensorimotor activity.

two daughter cells and, thus, cell fate (*Lancaster and Knoblich, 2012*). During normal cortical development, cleavage plane orientation is close to vertical relative to the VZ surface (60–90°). Disruption of mitotic spindle positioning can cause imbalance in IP and neuron production, which can affect cerebral cortex expansion (*Lancaster and Knoblich, 2012*; *Shitamukai and Matsuzaki, 2012*). Thus, we analyzed cleavage plane orientation in apical mitotic NPCs of control and *Smc5* cKO cortices at E16.5. Depletion of SMC5 in NPCs caused a significant increase in the number of anaphase cells with oblique division axis (<60°) (*Figure 1—figure supplement 1J*).

Our observations demonstrate that SMC5 depletion in NPCs results in the disruption of genomic integrity, abnormal mitosis, and cell death (*Figure 1D–I* and *Figure 1—figure supplement 1C–J*). Elimination of NPCs during neurodevelopment can cause decreased cortex size and lead to microcephaly (*Insolera et al., 2014*; *Lee et al., 2012*; *Marthiens et al., 2013*; *Martin et al., 2016*; *Mullegama et al., 2017*). Therefore, we investigated the *Smc5* cKO phenotype in newborn and adult brains. Tissue-specific depletion of SMC5 did not affect overall embryonic development and mice survived to adulthood without additional morphological abnormalities. The difference in brain weight and cortex size in *Smc5* cKO mediated by *Nestin-Cre* was significant at postnatal day 0/1 (P0/1) and became more prominent in adults (P55) (*Figure 1J–M* and *Figure 1—figure supplement 2A–D*). *Emx1-Cre*-mediated *Smc5* cKO also led to reduced brain weight and cortex size in newborn and adult mice compared to littermate controls (*Figure 1N,O* and *Figure 1—figure supplement 2E–L*).

Reduced mammalian cortex size can significantly affect behavioral performance (*Leingärtner et al., 2007*). To investigate if smaller cortex size affected sensorimotor activity in *Smc5* cKO mice, we performed three longitudinal sensorimotor assays (*Figure 1—figure supplement 3A–D*). The adhesive patch test allows for objective evaluation of sensory and motor activity, which is commonly used as an assessment following brain injury (*Bouet et al., 2009*; *Fleming et al., 2013*). This test consists of applying an adhesive patch on the mouse hind paw, and recording time spent to sense and remove the adhesive (*Bouet et al., 2009*). We observed a significant delay in the time taken to contact the adhesive in *Smc5* cKO mice compared to control mice, which is evidence of impaired sensory activity (*Figure 1—figure supplement 3A*). The inverted screen test measures the time spent upside down on the wire mesh screen and is indicative of muscle strength in all four limbs and brain motor activity (*Deacon, 2013*; *Grady et al., 2006*). *Smc5* cKO mice underperformed in this test compared to control mice (*Figure 1—figure supplement 3B*). The cylinder test allows for evaluation of exploratory behavior in a new environment, when mice rear and use their forelimbs against glass cylinder walls to support the body (*Fleming et al., 2013*; *Magno et al., 2019*). *Smc5* cKO mice displayed less forelimb use during vertical exploration compared to controls, as demonstrated by the elevated ratio of rears with no paws touching the cylinder wall and the reduced ratio of rears with two paws touching the cylinder wall (*Figure 1—figure supplement 3C, D*). Collectively, these behavioral tests revealed that the reduced cortex size mediated by SMC5 depletion results in impaired sensorimotor function in mice. Following conclusion of the sensorimotor assays, the aged adult (P293/295) *Smc5* cKO brains were assessed and showed reduced cortex size and thickness, as well as decreased brain weight, compared to the control mice (*Figure 1—figure supplement 3E–H*).

## Apoptosis of NPCs in *Smc5* cKO mice is mediated by p53 and CHEK2

We previously showed that *Smc5* cKO mouse embryonic fibroblasts (MEFs) accumulate RAD51 foci in the presence of replicative stress (*Gaddipati et al., 2019*). RAD51 nucleoprotein filament formation at stalled replication forks and on damaged DNA is critical for fork reversal and homology search and repair, respectively (*Quinet et al., 2017*; *Sullivan and Bernstein, 2018*; *Wright et al., 2018*). As NPCs are highly proliferative and have elevated propensity for endogenous replication stress and DNA damage, we assessed RAD51 levels during neurodevelopment (*Harley et al., 2016*;

*McKinnon, 2017*; *O'Driscoll, 2017*; *Reynolds et al., 2017*). Analysis of E16.5 embryonic tissues revealed upregulation of RAD51 in the neural cortex of *Smc5* cKO mice (*Figure 2A* and *Figure 2— figure supplement 1A*). DDR is controlled by ATM and ATR kinases, and in cases when DNA repair

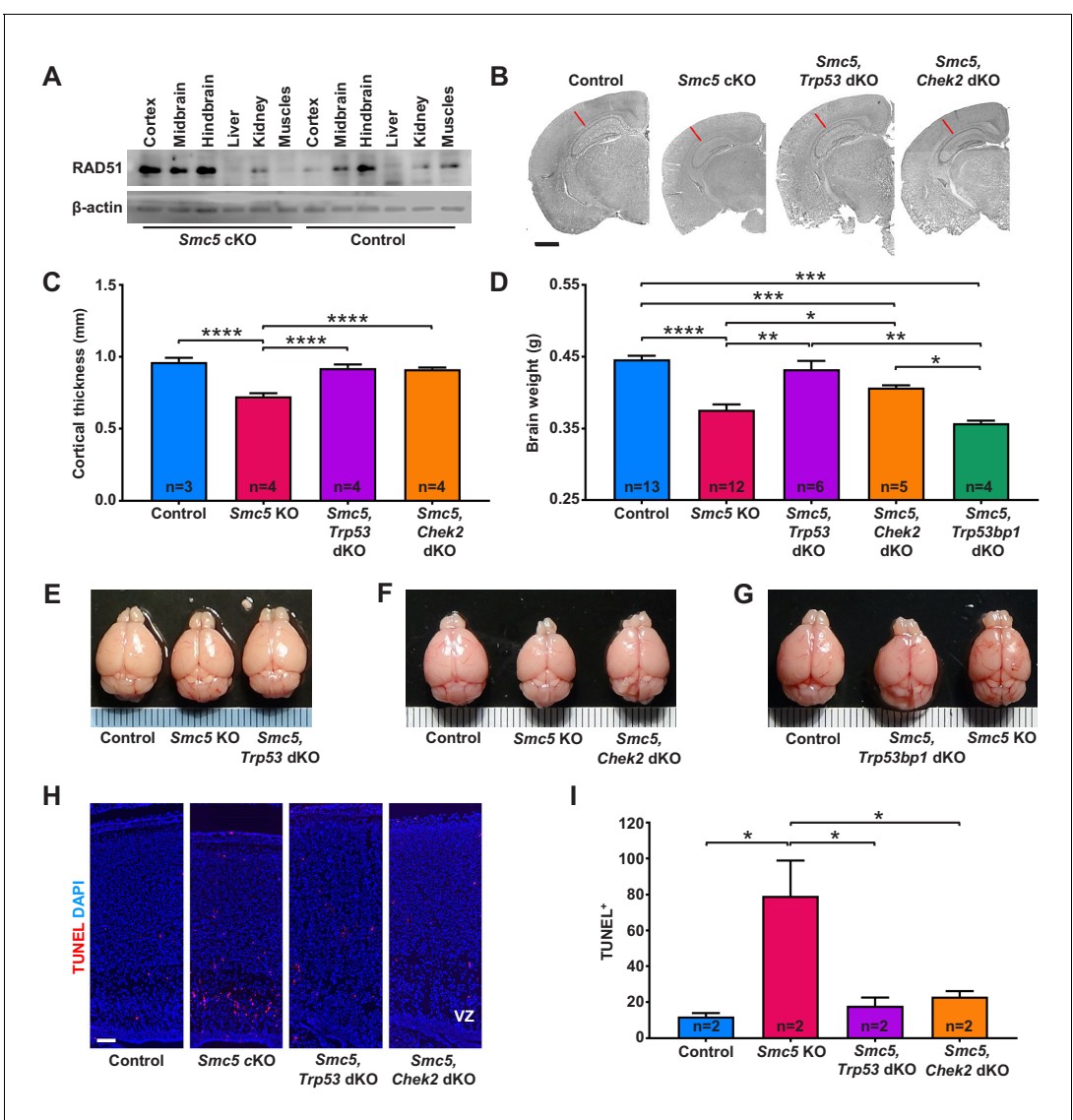

**Figure 2.** Apoptosis of neural progenitor cells in *Smc5* conditional knockout (cKO) mice is mediated by p53 and CHEK2 pathways. (**A**) Western blot analysis of RAD51 levels in control and *Smc5* cKO (*Nestin-Cre*) E16.5 tissues. β-actin was used as a loading control (n = 2). (**B**) Representative hematoxylin and eosin-stained control; *Smc5* cKO; *Smc5, Trp53* double knockout (dKO); and *Smc5, Chek2* dKO (*Emx1-Cre*) P55 coronal brain sections. Red line represents cortical thickness of *Smc5* cKO. Scale bar: 1000 μm. (**C**) Quantification of cortical thickness (mm) in control (n = 3); *Smc5* cKO (n = 4); *Smc5, Trp53* dKO (n = 4); and *Smc5, Chek2* dKO (n = 4) (*Emx1-Cre*) P54-57 brains. Data represent mean ± S.E.M. Unpaired two-tailed Mann–Whitney test, p-values are shown in *Supplementary file 3*. (**D**) Quantification of control (n = 13); *Smc5* cKO (n = 12); *Smc5, Trp53* dKO (n = 6); *Smc5, Chek2* dKO (n = 5); and *Smc5, 53* bp1 dKO (n = 4) (*Emx1-Cre*) P55-57 brain weight. Data represent mean ± S.E.M. Unpaired two-tailed Mann–Whitney test, p-values are shown in *Supplementary file 3*. (**E–G**) Representative images of (**E**) control; *Smc5* cKO; and *Smc5, Trp53* dKO; (**F**) control; *Smc5* cKO; and *Smc5 Chek2* dKO; and (**G**) control; *Smc5, Trp53bp1* dKO; and *Smc5* cKO (*Emx1-Cre*) P55 brains. Metric ruler is provided for scale. (**H**) Representative images of TUNEL+ nuclei (red) in control; *Smc5* cKO; *Smc5, Trp53* dKO; and *Smc5, Chek2* dKO (*Emx1-Cre*) P0/1 sagittal brain sections prepared from heads; DAPI (blue). Column width: 500 μm, scale bar: 100 μm. (**I**) Quantification of the number of TUNEL+ nuclei within 800 μm columns in brain sections related to (**H**). Data represent mean ± S.E.M. (control animals n = 2; *Smc5* cKO animals n = 2; *Smc5, Trp53* dKO animals n = 2; *Smc5, Chek2* dKO animals n = 2). Unpaired two-tailed Mann–Whitney test, p-values are shown in *Supplementary file 3*.

The online version of this article includes the following figure supplement(s) for figure 2:

**Figure supplement 1.** Cortical size in *Smc5* conditional knockout (cKO) mice can be restored by inhibition of p53 and CHEK2 pathways.

is not possible, these kinases activate p53 apoptotic signaling (*Enriquez-Rios et al., 2017*; *Maréchal and Zou, 2013*; *Sengupta and Harris, 2005*). We have previously demonstrated that p53 is upregulated and activated in MEFs and mESCs following *Smc5* cKO (*Gaddipati et al., 2019*; *Pryzhkova and Jordan, 2016*). Similarly, we observe p53 activation in the progenitor zone of E16.5 *Smc5* cKO cortices (*Figure 2—figure supplement 1B*). Therefore, we explored whether knockout of *Trp53* in the mouse cortex can alleviate the *Smc5* cKO phenotype. Brain weight was not affected in mice that were *Trp*53 KO alone (*Figure 2—figure supplement 1C*). The analysis of *Smc5*, *Trp53* double knockout (dKO) adult mice revealed that the brain weight, cortex area, and cortical thickness size were comparable to littermate controls (*Figure 2B–E* and *Figure 2—figure supplement 1D*). However, significant differences were observed compared to *Smc5* cKO mouse brain parameters (*Figure 2B–E* and *Figure 2—figure supplement 1D*). These data confirm that *Smc5* cKO in the mouse brain causes a p53-dependent apoptosis of neural cells.

Studies in *Drosophila*, budding yeast, and fission yeast have implied that aberrancies in SMC5/6 complex functions can result in the upregulation of a CHEK2 kinase-mediated DDR (*Ampatzidou et al., 2006*; *Pebernard et al., 2008b*; *Torres-Rosell et al., 2005*; *Tran et al., 2016*; *Winczura et al., 2019*). We reasoned that if NPC apoptosis in *Smc5* cKO brain is mediated by CHEK2 signaling pathway, *Chek2* knockout would rescue the reduced cortex size in *Smc5* cKO mice. Indeed, adult *Smc5*, *Chek2* dKO mouse brain weight, cortical thickness, and cortex area were significantly larger than *Smc5* cKO mice (*Figure 2B–D,F* and *Figure 2—figure supplement 1D*). However, the brain weight and cortex area were not restored to levels equivalent to the control or *Chek2* KO mice (*Figure 2B,D,F* and *Figure 2—figure supplement 1C,D*). This is likely because the p53-dependent apoptosis pathway is functional in *Smc5*, *Chek2* dKO mice, and defects that arise due to *Smc5* mutation still result in a degree of cell death. Taken together, these data suggest that abrogation of SMC5/6 functions results in DNA damage induced by DSBs and p53-mediated apoptosis likely through the ATM-CHEK2-p53 DDR signaling pathway.

A previous study in budding yeast suggested that the CHEK2 (Rad53) activation observed following SMC5/6 depletion was dependent on p53 binding protein 1 (53BP1) homolog, Rad9 (*Torres-Rosell et al., 2005*). 53BP1 interacts with ATM and plays a role in ATM and CHEK2 activation (*Mirza-Aghazadeh-Attari et al., 2019*). Earlier studies in 53BP1-deficient mice suggested that 53BP1 functions in DDR are regulated by ATM, and 53BP1 acts upstream of CHEK2 (*Ward et al., 2003*). Thus, we analyzed *Smc5*, *Trp53bp1* dKO adult mice and revealed that the brain weight and cortex size were not restored (*Figure 2D,G* and *Figure 2—figure supplement 1D*). These results demonstrate that the loss of SMC5/6 complex functions induces a DDR independent of or epistatic with 53BP1-mediated signaling.

Next, we investigated if the loss of p53 or CHEK2 can alleviate apoptosis in SMC5-deficient NPCs. In contrast to *Smc5* cKO, increased DNA fragmentation was not evident in *Smc5*, *Trp53* dKO and *Smc5*, *Chek2* dKO P0/1 cortices (*Figure 2H,I* and *Figure 2—figure supplement 1G*). From analysis of P0/1 sagittal head cross-sections, we observed rescue of the cortex thickness defect caused by *Smc5* cKO when *Trp53* or *Chek2* are also mutated (*Figure 2—figure supplement 1E,F*).

## SMC5 loss causes abnormal cortical development

Congenital microcephaly is often associated with abnormal cortical development due to NPC and cortical layer-specific neuron loss and their ectopic localization (*Insolera et al., 2014*; *Jiang and Nardelli, 2016*). Depletion of SMC5 in the developing mouse cortex caused the displacement of apical NPCs marked by SOX2 and PAX6 into the area outside VZ (*Figure 3A–F* and *Figure 3—figure supplement 1A,B*; *Englund et al., 2005*; *Ferri et al., 2004*; *Sansom et al., 2009*). This tendency was even more pronounced in *Smc5*, *Trp53* dKO and *Smc5*, *Chek2* dKO cortices (*Figure 3A–F* and *Figure 3—figure supplement 1A,B*). Interestingly, we observed an increase in the total number of SOX2+ and PAX6+ NPCs in both *Smc5*, *Trp53* dKO and *Smc5*, *Chek2* dKO compared to the control (*Figure 3A–F* and *Figure 3—figure supplement 1A,B*). The increase in NPCs and their ectopic localization may be due to a compensatory mechanism in response to the mitotic defects and apoptosis observed in SMC5-depleted NPCs, which may lead to perturbed differentiation (*Gaitanou et al., 2019*; *Lakomá et al., 2011*; *Savchenko et al., 2017*). In contrast to apical NPCs, we observed a significant decrease in the number of TBR2+ IPs located in the SVZ of *Smc5* cKO cortices compared to control (*Figure 3G–I*; *Arnold et al., 2008*; *Englund et al., 2005*). The cortical area occupied by TBR2+ progenitors in *Smc5*, *Trp53* dKO and *Smc5*, *Chek2* dKO cortices was increased compared to

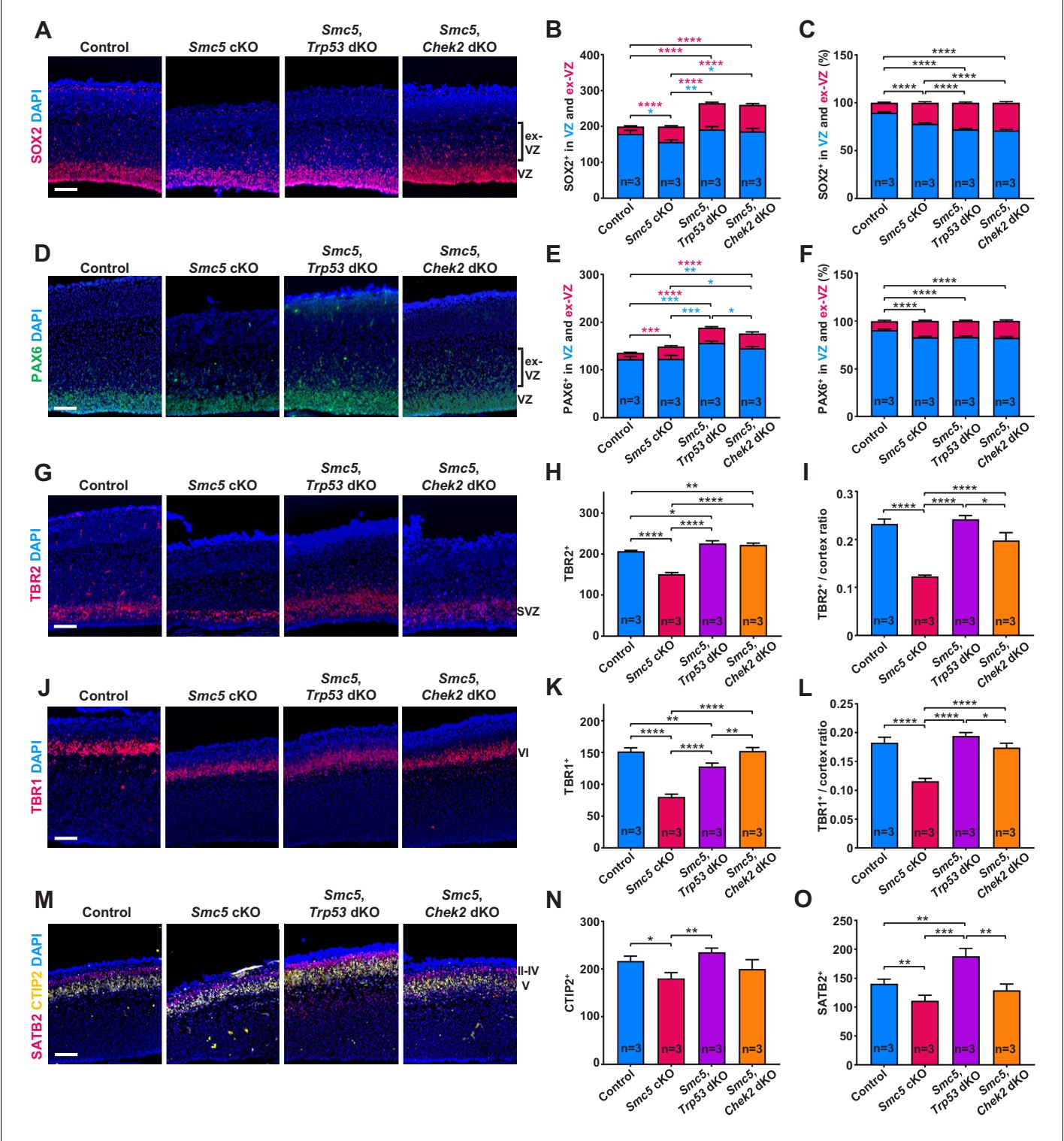

**Figure 3.** SMC5 loss causes abnormal cortical development. Representative immunostaining of control; *Smc5* conditional knockout (cKO); *Smc5, Trp53* double knockout (dKO) and *Smc5, Chek2* dKO (*Emx1-Cre*) E16.5 sagittal brain sections and quantification for: (A) SOX2 (red) (ventricular zone [VZ]), DAPI (blue). Bracket on the right shows extra-VZ (ex-VZ). (B) Quantification of SOX2+ cells in 150 μm columns in the VZ and extra-VZ from brain sections related to (A). (C) Percentage of SOX2+ cells in the VZ and extra-VZ within 150 μm columns in brain sections related to (A). (D) PAX6 (green) (VZ); DAPI (blue). Bracket on the right shows extra-VZ (ex-VZ). (E) Quantification of PAX6+ cells within 150 μm columns in the VZ and extra-VZ in brain sections related to (D). (F) Percentage of PAX6+ cells in the VZ and extra-VZ within 150 μm columns in brain sections related to (D). (G) TBR2 (red) (SVZ); DAPI

*Figure 3 continued on next page*

Figure 3 continued

(blue). (H) Quantification of TBR2+ cells within 150 µm columns in the SVZ in brain sections related to (G). (I) Quantification of TBR2-stained area thickness to cortical thickness ratio in brain sections related to (G). (J) TBR1 (red) (layer VI); DAPI (blue). (K) Quantification of TBR1+ cells within 150 µm columns in brain sections related to (J). (L) Quantification of TBR1-stained area thickness to cortical thickness ratio in brain sections related to (J). (M) CTIP2 (yellow) (layer V) and SATB2 (red) (layer II-V); DAPI (blue). (N) Quantification of CTIP+ cells within 300 µm columns in brain sections related to (M). (O) Quantification of SATB2+ cells within 300 µm columns in brain sections related to (M). For images A, D, G, J, and M column width: 500 µm, scale bar: 100 µm. For all graphs data represent mean ± S.E.M. Control animals n = 3; Smc5 cKO animals n = 3; Smc5, Trp53 dKO animals n = 3; Smc5, Chek2 dKO animals n = 3. For all graphs except C and F p-values were determined using unpaired two-tailed Mann–Whitney test. Pearson's chi-squared test with Yates' continuity correction was applied to determine p-value for graphs C and F. All p-values are shown in *Supplementary file 3*. The online version of this article includes the following figure supplement(s) for figure 3:

**Figure supplement 1.** Representative staining of E16.5 brains for cortical layer markers.

the control (*Figure 3G–I* and *Figure 3—figure supplement 1C*). These results demonstrate that *Smc5* cKO leads to ectopic localization of NPCs and significant loss of IPs. Mutation of *Trp53* or *Chek2* in conjunction with *Smc5* cKO alleviates the loss of IPs but exacerbates NPC ectopic localization.

We also investigated whether the lack of SMC5 affects the development of cortical layers. We observed a significant decrease in TBR1+ (layer VI), CTIP2+ (layer V), and SATB2+ (layers II–V) cells in *Smc5* cKO cortices compared to control (*Figure 3J–O* and *Figure 3—figure supplement 1D,E*). Assessment of the *Smc5, Trp53* dKO and *Smc5, Chek2* dKO cortices demonstrated that the development of early- and late-born neuron populations were restored (*Figure 3J–O* and *Figure 3—figure supplement 1D,E*).

Taken together, SMC5 depletion causes abnormal positioning of apical NPCs and reduction in the populations of IPs, as well as deep and upper cortical layer neurons, suggesting a global impact on cortical development. We also provide evidence that absence of p53 and CHEK2 prevents the loss of IPs and restores cortical layer-specific neuron numbers. These results imply that underlying reasons of affected corticogenesis are likely associated with functionality of apical NPCs. The fact that we do not observe a decrease in apical progenitors despite increased apoptosis in *Smc5* cKO cortices also implies impaired differentiation capability. Self-renewing NPCs have extended S-phase compared to progenitors committed to neuron production and can be subject to increased DNA replication stress (*Arai et al., 2011*; *Lavado et al., 2018*). Therefore, we next investigated DNA replication processes following depletion of SMC5.

## SMC5-deficient NPCs exhibit DNA replication stress

To investigate whether NPCs accumulated at specific stage of the cell cycle, we quantified proliferative Ki67+ cells, mitotic cells (phospho-histone H3 at serine 10, H3P+) and cells undergoing DNA replication (5-chloro-2'-deoxyuridine incorporation, CldU+) within the E16.5 cortex. *Smc5* cKO cortical sections had 20.2% less Ki67+ proliferating and 35.7% less H3P+ mitotic cells compared to control (*Figure 4A–D* and *Figure 4—figure supplement 1A–D*). We evaluated cells undergoing DNA synthesis by performing a 4-hr CldU pulse and analyzing the populations of CldU+ and H3P+ cells (*Figure 4E–G* and *Figure 4—figure supplement 1E,F*). We observed a 34.7% decrease in CldU+ populations in *Smc5* cKO cortices compared to control (*Figure 4E,F* and *Figure 4—figure supplement 1E,F*). The higher level of reduction in CldU+ and H3P+ populations compared to the reduction in total number of Ki67+ proliferating cells in *Smc5* cKO cortices indicates that there is an accumulation of cells in G1 phase of the cell cycle that are blocked from entering S phase (*Figure 4B,C,F,G*). The accumulation of cells in G1 phase may be a consequence of the DNA bridges and lagging chromosomes observed in *Smc5* cKO NPCs at anaphase (*Figure 1H,I*). CHEK2 is the primary checkpoint kinase during G1 phase that mediates p53 stabilization and cell cycle arrest (*Shaltiel et al., 2015*). These observations complement the fact that the *Smc5* cKO microcephaly phenotype is rescued by mutation of *Chek2* or *Trp53* (*Figures 2* and *3*, *Figure 2—figure supplement 1*, and *Figure 3—figure supplement 1*).

From our in vivo CldU pulse experiments, we observed NPCs with extensive incorporation of CldU that were undergoing aberrant chromosome segregation (*Figure 4H* and *Figure 4—figure supplement 1G*). Further, we observed accumulation of two intensely stained H3P+ chromatin foci on both sides of the segregating chromosomes that also contained CldU (*Figure 4H*). However, no

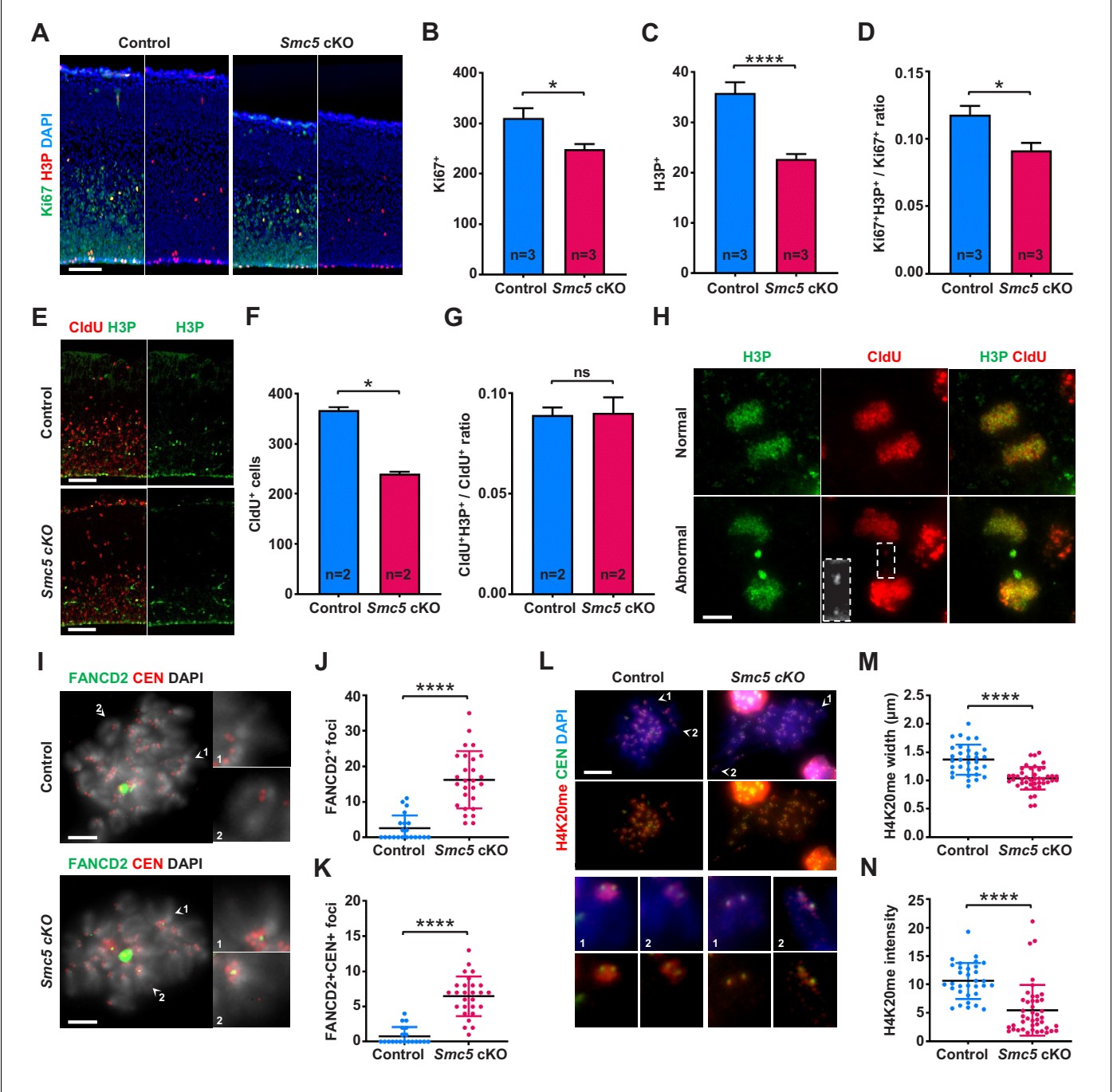

**Figure 4.** DNA replication stress is a possible reason for SMC5-deficient neural progenitor cell death. (**A**) Representative Ki67 (green) and H3P (red) staining of control and *Smc5* conditional knockout (cKO; *Emx1-Cre*) E16.5 sagittal cortical sections, DAPI (blue). Column width: 250 μm, scale bar: 100 μm. (**B**) Quantification of Ki67+ cells within 300 μm columns in brain sections related to (**A**). Data represent mean ± S.E.M. (control animals n = 3, *Smc5* cKO animals n = 3, see ***Supplementary file 3*** for details). Unpaired two-tailed Mann–Whitney test, *p=0.0443. (**C**) Quantification of H3P+ cells within 300 μm columns in brain sections related to (**A**). Data represent mean ± S.E.M. (control animals n = 3, *Smc5* cKO animals n = 3, see ***Supplementary file 3*** for details). Unpaired two-tailed Mann–Whitney test, ****p<0.0001. (**D**) Quantification of H3P+Ki67+/Ki67+ cell ratio within 300 μm columns in brain sections related to (**A**). Data represent mean ± S.E.M. (control animals n = 3, *Smc5* cKO animals n = 3, see ***Supplementary file 3*** for details). Unpaired two-tailed Mann–Whitney test, *p=0.0130. (**E**) Representative CldU (red) and H3P (green) staining of control and *Smc5* cKO (*Emx1-Cre*) E16.5 sagittal brain sections. Column width 300 μm, scale bar: 100 μm. (**F**) Quantification of CldU+ cells within 300 μm columns in brain sections related to (**E**). Data represent mean ± S.E.M. (control animals n = 2; *Smc5* cKO animals n = 2, see ***Supplementary file 3*** for details). Unpaired two-tailed Mann–Whitney test, *p=0.0286. (**G**) Quantification of CldU+H3P+/CldU+ cell ratio within 300 μm columns in brain sections related to (**E**). Data represent mean ± S.E.M. (control animals n = 2; *Smc5* cKO animals n = 2, see ***Supplementary file 3*** for details). Unpaired two-tailed Mann–Whitney test, ns, not significant. (**H**)

*Figure 4 continued on next page*

*Figure 4 continued*

Representative images of normal and abnormal anaphase cells from E16.5 *Smc5* cKO (*Emx1-Cre*) cortices stained with CldU (red) and H3P (green). Dashed rectangle outlines CldU+ chromatin bridge. Scale bar: 5 µm. (I) Representative FANCD2 (green) and CEN (red) staining of control and *Smc5* cKO (*Nestin-Cre*) E16.5 cortical cell spreads; DAPI (white). Insets depict magnified individual chromosomes. Scale bar: 5 µm. (J) Quantification of average number of FANCD2 foci in cortical cells related to (I). Data represent mean ± S.D. (control animals n = 2; control cells n = 21; *Smc5* cKO animals n = 2; *Smc5* cKO cells n = 27). Unpaired two-tailed Mann–Whitney test, ****p<0.0001. (K) Quantification of average number of FANCD2+CEN+ foci in cortical cells related to (I). Data represent mean ± S.D. (control animals n = 2; control cells n = 20; *Smc5* cKO animals n = 2; *Smc5* cKO cells n = 27). Unpaired two-tailed Mann–Whitney test, ****p<0.0001. (L) Representative H4K20me (red) and centromere (CEN) (green) staining of control and *Smc5* cKO (*Nestin-Cre*) E16.5 cortical cell chromosome spreads; DAPI (blue). Insets depict magnified individual chromosomes. Scale bar: 10 µm. (M) Quantification of average width of pericentromeric H4K20me signal in cortical cells related to (L). Data represent mean ± S.D. (control animals n = 2; control cells n = 32; *Smc5* cKO animals n = 2; *Smc5* cKO cells n = 43). Unpaired two-tailed Mann–Whitney test, ****p<0.0001. (N) Quantification of average intensity of pericentromeric H4K20me signal in cortical cells related to (L). Data represent mean ± S.D. (control animals n = 2; control cells n = 32; *Smc5* cKO animals n = 2; *Smc5* cKO cells n = 43). Unpaired two-tailed Mann–Whitney test, ****p<0.0001.
The online version of this article includes the following figure supplement(s) for figure 4:

**Figure supplement 1.** Representative staining of E16.5 cortices for proliferative cell markers.

CldU signal was observed between either of the two intensely stained H3P+ foci. As we observed NPCs within the *Smc5* cKO cortex with aberrant DNA bridges during anaphase (*Figure 1H,I*), we postulated that these events may be due to an inability to complete DNA replication prior to chromosome segregation. The detection of ultrafine DNA bridges formed by under-replicated DNA on histological sections is technically challenging, and our data require further support. Thus, we next focused on the analysis of chromosome spreads prepared using cells dissociated from E16.5 cortices.

Segregation of under-replicated chromosomes is generally coupled with a DNA replication stress response (*Berti and Vindigni, 2016*). Fanconi anemia (FA) components, such as FANCD2, are known to accumulate at sites of DNA replication stress and, particularly, at common fragile sites (*Datta and Brosh Jr., 2019*). Furthermore, late-replicating regions, such as pericentromeric heterochromatin, are additional sites prone to DNA replication stress (*Mendez-Bermudez et al., 2018*; *Saksouk et al., 2015*). Therefore, we assessed FANCD2 signal on chromosome spreads of prometaphase stage NPCs isolated from control and *Smc5* cKO E16.5 cortices. In all chromosome spreads prepared from control and *Smc5* cKO NPCs, we observed a large FANCD2 signal, which likely corresponds to the centrosome, as previously reported for FA proteins, including FANCD2 (*Nalepa et al., 2013*). In addition, we observed increased number of FANCD2+ foci in *Smc5* cKO compared to control cells, particularly at pericentromeric regions (*Figure 4I–K*). The number of FANCD2+ foci in *Smc5* cKO cells was increased by 6.5-fold compared to control, while pericentromeric FANCD2+ foci were increased by 8.1-fold (*Figure 4J,K*).

We have previously shown that SMC5/6 components accumulate at pericentromeric heterochromatin regions of chromosomes (*Hwang et al., 2018*; *Hwang et al., 2017*; *Pryzhkova and Jordan, 2016*). Therefore, we wondered whether the absence of SMC5/6 may disrupt the heterochromatic features of this region of the genome. Previous studies have indicated that histone H4 lysine 20 trimethylation (H4K20me3) is enriched at pericentromeric heterochromatin (*Gonzalo et al., 2005*; *Schotta et al., 2004*). H4K20 methylation regulates heterochromatin compaction during mitosis and is particularly important for regulating DNA replication and maintaining genome integrity (*Jørgensen et al., 2013*; *Saksouk et al., 2015*; *Schotta et al., 2008*; *Shoaib et al., 2018*). Assessment of chromosome spreads prepared from control and *Smc5* cKO E16.5 cortical cells revealed a decrease in average width and intensity of pericentromeric H4K20 signal by 1.3- and 2.0-fold, respectively, in *Smc5* cKO cells compared to control (*Figure 4L–N* and *Figure 4—figure supplement 1H*). H4K20 methylation is reduced in response to DNA damage allowing for decompaction of chromatin structure, DNA synthesis, and efficient DNA repair (*Jørgensen et al., 2013*). Reduction in H4K20 methylation is also associated with increased replication origin licensing (*Shoaib et al., 2018*). Therefore, the reduction in methylated pericentromeric H4K20 in *Smc5* cKO cells may reflect under-replicated DNA and increased DNA damage.

## Perturbed proliferation of SMC5-depleted mESCs can be alleviated by CHEK2 or p53 inhibition

Similar to NPCs, mESCs are rapidly proliferating stem cells subject to high levels of replication stress (*Ahuja et al., 2016*). To further investigate potential causes of genome instability, we assessed SMC5-depleted mESCs in detail. In a previous study, we showed that tamoxifen-induced knockout of *Smc5* in mESCs causes a dramatic decrease in cell growth and p53 signaling is upregulated (*Pryzhkova and Jordan, 2016*). In this study, we utilized a mESC line harboring the AID system, which allows for rapid depletion of SMC5 upon addition of indole-3-acetic acid (IAA) (hereafter referred to as Smc5-AID mESCs) (*Figure 5A*; *Natsume et al., 2016*; *Nishimura et al., 2009*; *Pryzhkova et al., 2020*). We observed the reduction of SMC5 levels as early as 30 min after IAA addition, with robust depletion occurring after 1 hr of IAA treatment (*Figure 5B* and *Figure 5—figure supplement 1A*).

Immunoblotting analysis showed that IAA treatment triggered upregulation of acetyl-p53 (K379) and phospho-p53 (S389) by 2 hr and 48 hr, respectively (*Figure 5C,D*). This is consistent with the fact that Smc5-AID mESCs undergo severe cell growth perturbations within 48 hr after SMC5 depletion (*Figure 5E*).

In addition, we performed immunofluorescence analysis of phospho-p53 (p-p53) (S389) expression at various cell cycle stages (*Figure 5—figure supplement 1D*). To define proliferative cells, we performed immunostaining for Ki67 and H3P at the 48 hr time point. Nearly 100% of control and IAA-treated mESCs were Ki67+, and we never observed p-p53 signal in Ki67- cells. Previous reports demonstrate that H3P is first detectable in mESCs during S phase and reaches a maximum during mitosis (*Mallm and Rippe, 2015*; *Pryzhkova and Jordan, 2016*). H3P is not detectable in G1 phase mESCs. Greater than 90% of S phase mESCs exhibit H3P foci, that become more prominent by G2, colocalizing with the pericentromeric heterochromatin. By prophase H3P stains the entire chromatin of the cell. Thus, we defined the H3P- population as containing primarily G1 phase cells and classified the H3P+ population as non-mitotic (S/G2 phase), prophase or cells in other mitotic stages. We observed a global increase in the percentage of p-p53+ cells in both H3P+ and H3P- populations after 48 hr of IAA treatment. Specifically, 38.2% of G1 phase cells, 26.2% of S/G2 phase cells, and 40.8% of prophase cells were positive for p-p53 in the IAA-treated group compared to 11.9%, 11.8%, and 7.0% of the control in respective stages (*Figure 5—figure supplement 1E*). The relative abundance of p-p53+ cells in G1 and prophase populations compared to S/G2 cells suggests that p53 (S389) phosphorylation is most likely to occur in G1 phase or prophase upon SMC5 depletion. Further reaffirming this, the majority of p-p53+ cells were in G1 phase in the IAA-treated group, while the majority of p-p53+ cells in the control population were in S/G2 phase (*Figure 5—figure supplement 1F*). We also observed a 1.5-fold increase in the percentage of total G1 phase cells, a 1.5-fold decrease in the percentage of total S/G2 phase cells, and a 2.1-fold increase in the percentage of total prophase cells in the IAA-treated group compared to control (*Figure 5—figure supplement 1E*). The activation of a DDR during G1 or prophase, leading to activation of p53 and cell death, might explain the increase in proportion of cells in these cell cycle stages. These observations are consistent with the apparent accumulation of cells in G1 phase in *Smc5* cKO cortices (*Figure 4B, C,F,G*). Furthermore, the accumulation of cells in prophase recapitulates our previous findings in *Smc5* knockout mESCs (*Pryzhkova and Jordan, 2016*).

The *Smc5* cKO reduced cortex size can be alleviated by knockout of *Trp53* or *Chek2* (*Figures 2* and *3*, *Figure 2—figure supplement 1*, and *Figure 3—figure supplement 1*). CHEK1 and CHEK2 kinases are key transducers of DNA damage signaling that lead to DNA repair, p53-mediated cell cycle arrest, or p53-mediated apoptosis (*Shaltiel et al., 2015*; *Smith et al., 2010*). We tested whether the phenotype of SMC5-depletion in mESCs can be rescued by inhibition of CHEK1, CHEK2, or p53 (*Figure 5E* and *Figure 5—figure supplement 1B,C*). Chk2 inhibitor II and LY2603618 (LCI-1) are potent and selective inhibitors of CHEK2 and CHEK1, respectively (*Arienti et al., 2005*; *Dai et al., 2011*; *King et al., 2014*). Concurrent treatment with IAA and CHEK2 inhibitor restored cell numbers similar to the CHEK2 inhibitor-only control (*Figure 5E*). Similar results were obtained when mESCs were incubated with a commonly used p53 inhibitor, pifithrin-α, although potential off-target effects of this compound cannot be excluded (*Figure 5—figure supplement 1B*; *Sohn et al., 2009*; *Zhu et al., 2020*). In contrast, CHEK1 inhibition did not rescue cell growth defects after IAA treatment (*Figure 5—figure supplement 1C*). Collectively, these results

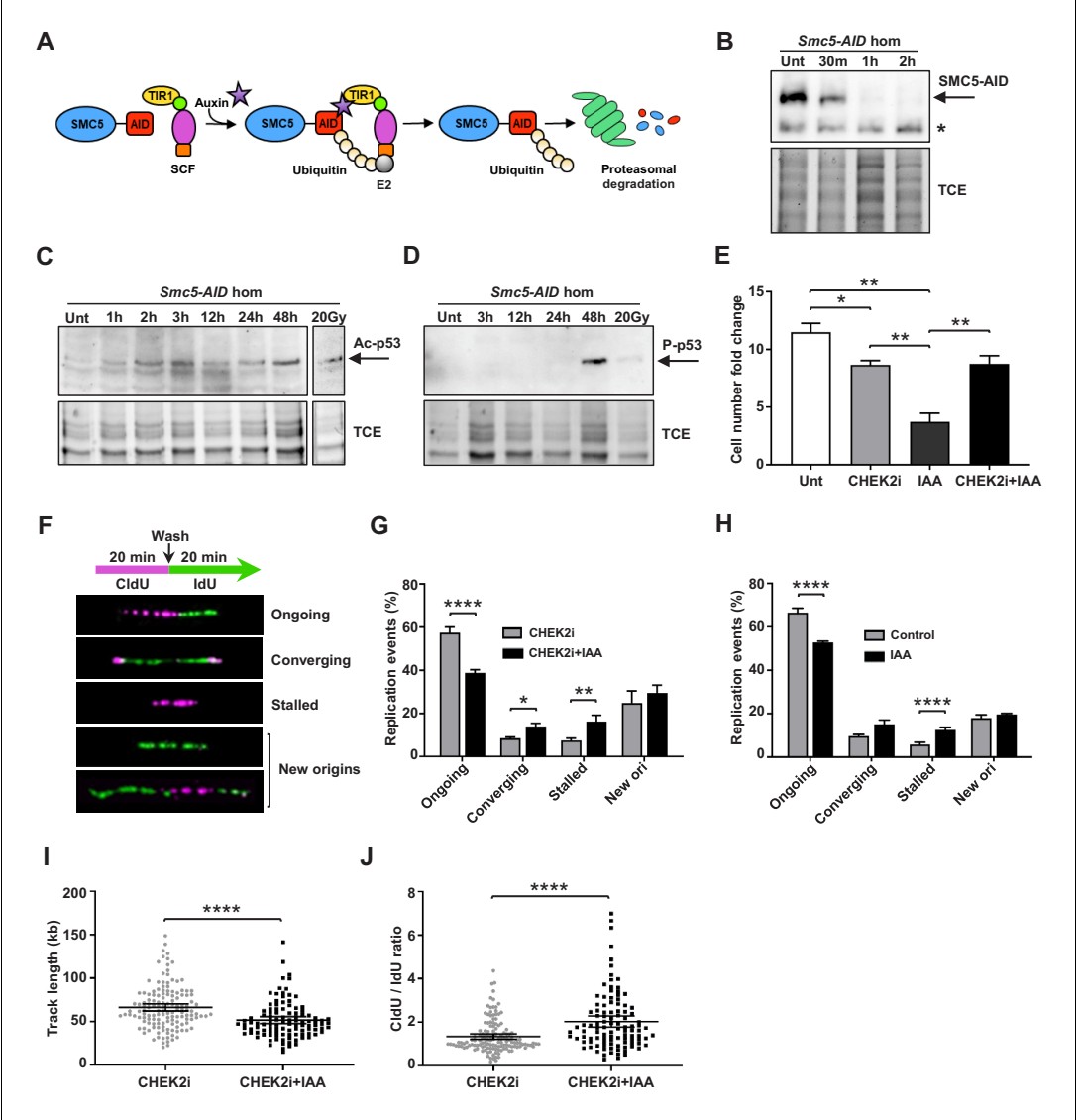

**Figure 5.** Depletion of SMC5 in mESCs perturbs cell growth and causes increased replication stress. (**A**) Schematic of AID system. Upon addition of IAA, SMC5-AID is ubiquitinated by the SKP1, CULLIN1, F-box (SCF) complex and degraded by the proteasome. (**B**) Western blot analysis of SMC5-AID protein levels over a time course of IAA (100 μM) treatment (Unt, untreated, 30 min, 1 hr, 2 hr) in *Smc5-AID* homozygous mESCs. Asterisk marks nonspecific band. 2,2,2-Trichloroethanol (TCE) was incorporated in the gel to visualize total loaded protein. (**C**) Western blot analysis of acetyl-p53 (K379) protein levels over a time course of IAA (100 μM) treatment (Unt, untreated, 1 hr, 2 hr, 3 hr, 12 hr, 24 hr, 48 hr) in *Smc5-AID* homozygous mESCs. A separate western blot of acetyl-p53 (K379) in *Smc5-AID* homozygous mESCs treated with 20 Gy of gamma-irradiation is shown on the right. 2,2,2-Trichloroethanol (TCE) was incorporated in the gel to visualize total loaded protein. (**D**) Western blot analysis of phospho-p53 (S389) protein levels over a time course of IAA (100 μM) treatment (Unt, untreated, 3 hr, 12 hr, 24 hr, 48 hr), and with 20 Gy of gamma-irradiation , in *Smc5-AID* homozygous mESCs. 2,2,2-Trichloroethanol (TCE) was incorporated in the gel to visualize total loaded protein. (**E**) *Smc5-AID* mESC number fold change over 48 hr of cell culture without treatment (Unt), and in the presence of IAA (100 μM), CHEK2 inhibitor (CHEK2i) (10 μM), or IAA and CHEK2 inhibitor (CHEK2i + IAA). Data represent mean ± S.E.M. (n = 3). Unpaired two-tailed t-test, untreated versus CHEK2i *p=0.0289, untreated versus IAA **p=0.0019; IAA versus CHEK2i **p=0.0042; IAA versus CHEK2i+IAA **p=0.0086 (see *Supplementary file 3* for details). (**F**) DNA labeling scheme and representative images of DNA fibers labeled with CldU (30 μM) (magenta) and IdU (250 μM) (green) depicting ongoing forks, double-stalled converging forks, stalled forks, and new origins. (**G**) Quantification of replication events in *Smc5-AID* mESCs treated for 12 hr with CHEK2i and IAA or CHEK2i alone. Data represent mean ± S.E.M. (CHEK2i condition: n = 354 fibers from three experiments, CHEK2i + IAA condition: n = 429 fibers from three experiments). Pearson's chi-squared test with Yates' continuity correction, *p=0.0436, **p=0.0025, ****p<0.0001 (see *Supplementary file 3* for details). (**H**) Quantification of replication events in control *Smc5-AID* mESCs or *Smc5-AID* mESCs treated for 12 hr with IAA. Data represent mean ± S.E.M. (control condition: n = 334 fibers from three experiments, IAA condition: n = 558 fibers from three experiments). Pearson's chi-squared test with Yates' continuity correction, ****p<0.0001 (see *Supplementary file 3* for details). (**I**) Quantification of DNA fiber track length in *Smc5-AID* mESCs treated for 12 hr with CHEK2i and IAA or CHEK2i alone. Data represent mean ± 95% C.I. (confidence interval) (CHEK2i condition: n = 142 fibers from three

*Figure 5 continued on next page*

*Figure 5 continued*

experiments, CHEK2i + IAA condition: n = 107 fibers from three experiments). Unpaired two-tailed Mann–Whitney test, ****p<0.0001. (J) Quantification of DNA fiber CldU/IdU ratio in *Smc5-AID* mESCs treated for 12 hr with CHEK2i and IAA or CHEK2i alone. Data represent mean ± 95% C.I. (CHEK2i condition: n = 142 fibers from three experiments, CHEK2i + IAA condition: n = 107 fibers from three experiments). Unpaired two-tailed Mann–Whitney test ****p<0.0001.

The online version of this article includes the following figure supplement(s) for figure 5:

**Figure supplement 1.** Depletion of SMC5 in mESCs perturbs cell growth and causes increased replication stress.

suggest that the main pathway triggering p53 activation, cell cycle arrest, and apoptosis upon SMC5 depletion in mESCs is CHEK2-mediated. This is consistent with our in vivo findings, which demonstrate that the *Smc5* cKO neurodevelopmental defects can be alleviated by mutation of *Trp53* or *Chek2* (*Figures 2* and *3*, *Figure 2—figure supplement 1*, and *Figure 3—figure supplement 1*).

## Depletion of SMC5 causes increased DNA replication stress

The CHEK2 pathway largely responds to DSBs, which are commonly formed during replication stress, the primary source of DNA damage in rapidly proliferating stem cells (*Ahuja et al., 2016*; *Zeman and Cimprich, 2014*). Based on studies using budding and fission yeast, it is proposed that the SMC5/6 complex is important for maintaining replication fork stability and avoiding accumulation of toxic recombination intermediates (*Aragón, 2018*). Therefore, we reasoned that perturbed DNA replication could be responsible for genome instability and cell death in SMC5-deficient stem cells. To assess the effect of SMC5 depletion on replication fork progression, we performed DNA fiber assays, in which mESCs were pulse-labeled with CldU and then IdU (*Figure 5F*). We found that SMC5 depletion in CHEK2-inhibited mESCs resulted in 1.5-fold decrease in ongoing unidirectional forks compared to the inhibitor-only control (*Figure 5G*). In addition, converging forks and stalled forks were increased by 1.6- and 2.2-fold, respectively, compared to control (*Figure 5G*). These observations suggest that replication stress is increased upon SMC5 depletion, perhaps due to an impaired ability to restart stalled replication forks or resolve replication intermediates. This may lead to licensing of nearby dormant replication origins, which can increase the prevalence of converging forks (*Blow and Ge, 2009*; *Merrick et al., 2004*). Increased converging fork frequency in SMC5-depleted cells could also be indicative of an inability to resolve joint molecules formed upon collision of two replication forks (*Dewar and Walter, 2017*). SMC5 depletion in p53-inhibited mESCs yielded similar results (*Figure 5—figure supplement 1G*). The same trend was observed in SMC5-depleted mESCs in the absence of inhibitors (*Figure 5H*). We also observed 1.3-fold reduced track length and 1.5-fold elevated CldU/IdU ratio in unidirectional dual-labeled forks upon SMC5 depletion with CHEK2 inhibition (*Figure 5I,J*). This further affirms that fork stalling is increased, and replication is perturbed in SMC5-depleted cells. The decrease in track length and increase in CldU/IdU ratio upon SMC5 depletion were also recapitulated in the absence of inhibitors (*Figure 5—figure supplement 1H,I*). However, the track length reduction, while significant, was not as pronounced in these conditions. The slightly less severe phenotypes in the absence of inhibitors could be attributed to increased death of SMC5-deficient cells when CHEK2 and p53 are active.

## SMC5 depletion leads to increased mitotic DNA synthesis

The prevalence of stalled and converging forks in SMC5-depleted mESCs suggests their inability to restart stalled forks or resolve replication intermediates (*Berti and Vindigni, 2016*). Unresolved replication intermediates could persist into mitosis, where they can be repaired via HDR (termed mitotic DNA synthesis, MiDAS) (*Mankouri et al., 2013*; *Minocherhomji et al., 2015*; *Sonneville et al., 2019*). MiDAS often occurs at common fragile sites, AT-rich DNA sequences that are particularly prone to replication stress and subsequent breakage during mitosis, as well as at late-replicating heterochromatin regions (*Fungtammasan et al., 2012*; *Glover et al., 2005*; *Mendez-Bermudez et al., 2018*; *Özer and Hickson, 2018*; *Saksouk et al., 2015*). Sites of MiDAS are generally accompanied by foci of FANCD2, which is required for DNA crosslink repair and regulation of MiDAS (*Bhowmick et al., 2016*; *Datta and Brosh Jr., 2019*; *Graber-Feesl et al., 2019*; *Okamoto et al., 2018*). Thus, we evaluated the frequency of MiDAS in synchronized mESCs during mitosis (*Figure 6*).

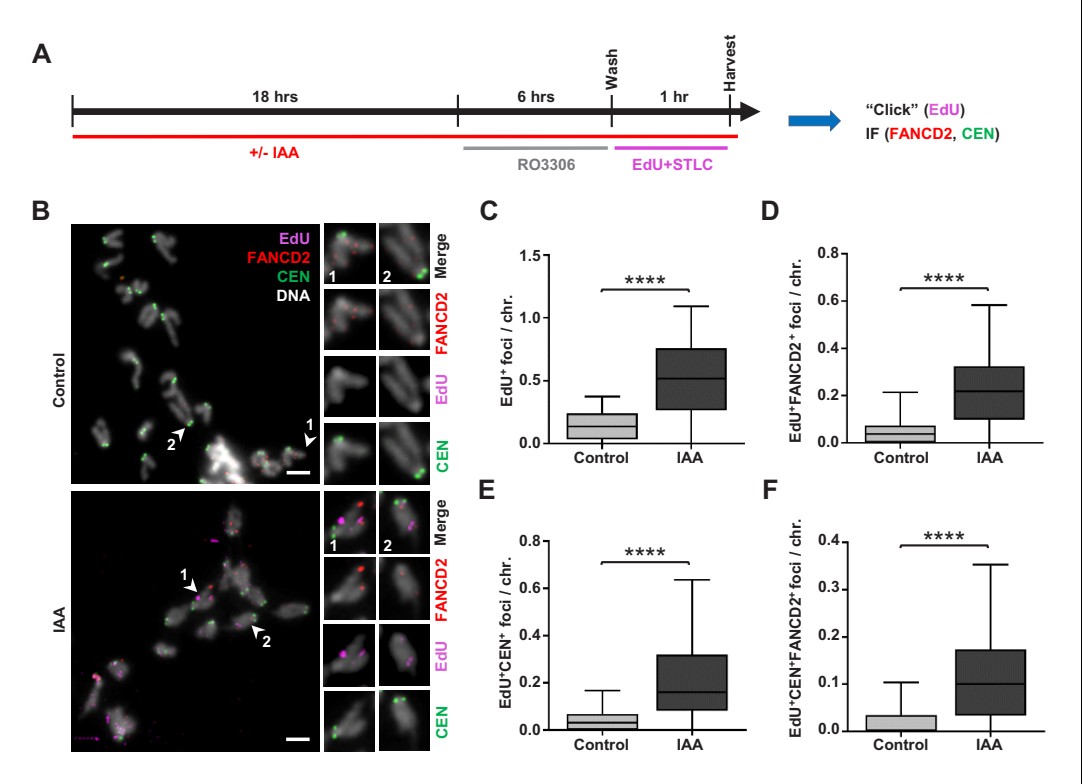

**Figure 6.** Replication stress in SMC5-depleted mESCs leads to MiDAS. (**A**) Schematic of IAA treatment, cell synchronization with RO3306 (8 µM) and STLC (10 µM), and labeling with EdU (10 µM) for assessment of MiDAS foci. (**B**) Representative images of EdU (magenta), FANCD2 (red), and centromere (CEN) (green) staining of metaphase chromosomes from control and IAA-treated *Smc5-AID* mESCs; DAPI (white). Insets depict magnified individual chromosomes with combined channels and separate channels. Scale bar: 5 µm. (**C**) Quantification of EdU+ foci per chromosome in control (n = 34) and IAA-treated (n = 37) mESCs. Data represent mean and range. Unpaired two-tailed Mann–Whitney test, ****p<0.0001. (**D**) Quantification of EdU+FANCD2+ foci per chromosome in control (n = 34) and IAA-treated (n = 37) mESCs. Data represent mean and range. Unpaired two-tailed Mann–Whitney test, ****p<0.0001. (**E**) Quantification of EdU+CEN+ foci per chromosome in control (n = 34) and IAA-treated (n = 37) mESCs. Data represent mean and range. Unpaired two-tailed Mann–Whitney test, ****p<0.0001. (**F**) Quantification of EdU+FANCD2+CEN+ foci per chromosome in control (n = 34) and IAA-treated (n = 37) mESCs. Data represent mean and range. Unpaired two-tailed Mann–Whitney test, ****p<0.0001.

The online version of this article includes the following figure supplement(s) for figure 6:

**Figure supplement 1.** Quantification of EdU+ and FANCD2+ foci in mESCs.

IAA-treated and untreated mESCs were arrested in G2 phase using the CDK1 inhibitor RO3306, released, and subsequently arrested in mitosis using the Eg5 inhibitor S-trityl-L-cysteine (STLC) in the presence of thymidine analogue EdU (*Figure 6A*). Higher frequency of MiDAS was observed in SMC5-depleted mESCs, indicated by EdU+ and dual EdU+FANCD2+ foci (*Figure 6B–D*). The mean number of EdU+ and dual EdU+FANCD2+ foci per chromosome was increased by 3.6- and 4.3-fold, respectively, in SMC5-depleted mESCs compared to control (*Figure 6B–D*). The percentage of EdU foci that did not co-localize with FANCD2 was 59.4% and 56.2%, respectively, for control and IAA-treated cells (*Figure 6—figure supplement 1A*). We also found that 81.3% and 64.5% of FANCD2 foci were EdU- in control and IAA-treated cells, respectively (*Figure 6—figure supplement 1B*). Thus, FANCD2 foci are less frequently found outside of MiDAS foci upon SMC5 depletion, consistent with the increased number of EdU+FANCD2+ foci seen in IAA-treated cells (*Figure 6D*). Loss of SMC5 also caused elevated MiDAS at pericentromeric regions, as indicated by dual EdU+CEN+ foci and triple EdU+FANCD2+CEN+ foci, which were increased by 4.7- and 6.2-fold, respectively (*Figure 6B,E,F*). To note, EdU foci positioning at telomeres and centromeres on the acrocentric end of mouse chromosomes is difficult to distinguish due to their proximity. Therefore, we also quantified the ratio of 'acrocentric' EdU foci (that is, foci present on the end of the chromosome containing the centromere) to 'non-acrocentric' EdU foci (foci present on the opposite chromosomal end). This ratio was 1.47 and 1.91 in control and IAA-treated mESCs, respectively (*Figure 6—figure*

supplement 1C). The enrichment of acrocentric compared to non-acrocentric foci suggests that features unique to the pericentromeric DNA may confer increased MiDAS susceptibility and complements our findings in *Smc5* cKO NPCs compared to control NPCs (*Figure 4I–K*).

Our data suggest that the depletion of SMC5 in stem cells results in increased DNA replication stress (*Figure 5F-J* and *Figure 5—figure supplement 1G-I*). This could lead to the formation of under-replicated DNA and replication intermediates that can persist into mitosis, particularly at sites prone to replication stress. Inability to resolve and repair these forms of DNA joint molecules can cause chromosome segregation errors, which we have observed in NPCs of *Smc5* cKO mice and mESCs (*Figure 1H,I*, *Figure 4H*, and *Figure 4—figure supplement 1G*).

## Discussion

### The SMC5/6 complex is critical for normal brain development and function

The SMC5/6 complex plays a crucial role in preserving genome integrity of somatic and stem cells (*Gallego-Paez et al., 2014*; *Pryzhkova and Jordan, 2016*; *Venegas et al., 2020*). Mutation of *NSMCE2* causes primordial dwarfism and primary congenital microcephaly in humans and promotes cancer development and premature aging in mice (*Jacome et al., 2015*; *Payne et al., 2014*). Our study provides the first insight toward understanding what neurodevelopmental processes are likely to be aberrant in humans harboring SMC5/6 mutations. The major neurodevelopmental defects we observed in our *Smc5* cKO model during embryogenesis were the displacement of apical NPCs into the cortical area outside the VZ and reduction in IPs and neurons of all cortical layers. However, the NPC population was not decreased by *Smc5* cKO. Therefore, our data suggest that a proportion of NPCs in the *Smc5* cKO cortex do not contribute to the development of IPs and fail to differentiate into layer-specific neurons. In mouse models, NPC genome instability often leads to lethality during development in utero or shortly after birth (*Katyal et al., 2014*; *Lee et al., 2012*; *McKinnon, 2017*; *Nishide and Hirano, 2014*; *Rosin et al., 2015*). In contrast, cKO of *Smc5* within the developing cortex does not affect animal survival into adulthood and can be used for longitudinal behavioral, sensorimotor, and neuronal activity analyses. We have shown that *Smc5* cKO mice have sensorimotor defects, and they may prove to be an instrumental model for more comprehensive studies in the future.

### SMC5/6 absence triggers activation of CHEK2- and p53-mediated apoptosis

It has been well documented that a major source of DNA damage during neurogenesis is DNA replication stress (*Lee et al., 2012*; *Magdalou et al., 2014*; *McKinnon, 2013*; *O'Driscoll, 2017*; *Zeman and Cimprich, 2014*). ATR kinase and its downstream substrate CHEK1 mediate a checkpoint response in the presence of aberrant replication fork structures containing ssDNA (*Zeman and Cimprich, 2014*). *Smc5* cKO during neurodevelopment did not result in accumulation of NPCs in S phase, which indicated that the ATR-CHEK1 S phase checkpoint pathway did not trigger cell cycle arrest. In contrast, we observed *Smc5* cKO NPCs undergoing mitosis with DNA bridges and lagging chromosomes at anaphase. Despite the chromosome segregation defects, *Smc5* cKO NPCs did not undergo mitotic delay (*Phan et al., 2020*). These data suggested that a DNA damage checkpoint was not being activated during G2 or M phase. DNA bridges can lead to the inheritance of DNA breaks in G1 daughter cells; therefore, it is likely that NPCs in G1 are subject to a DNA damage checkpoint response.

ATM together with CHEK2 are the primary DNA damage signaling proteins during G1 and result in the stabilization of p53 (*Shaltiel et al., 2015*). We demonstrated that p53- and CHEK2-mediated apoptosis were major drivers for the reduced cortex size following *Smc5* cKO in NPCs. The depletion of SMC5 in mESCs resulted in upregulation of phospho- and acetyl-p53, consistent with the phenotype observed in mice. We further showed that the acetylation of p53 occurs within 2 hr after IAA treatment, while p53 phosphorylation occurs significantly later, presumably after multiple rounds of cell division. Given that complete SMC5 depletion occurs only after 1–2 hr of IAA treatment, the upregulation of acetyl-p53 at this early time point reveals that a DDR occurs nearly immediately upon SMC5 depletion. By contrast, previous studies in human cancer cells demonstrated an increase

in unmodified p53 levels only within 2 days after SMC5/6 depletion by the AID system (*Venegas et al., 2020*). Immunofluorescence microscopy analyses demonstrated that p-p53 was most frequently present in G1 phase and prophase cells, suggesting that aberrancies caused by SMC5 depletion lead to a DDR during G1 and prophase. As we observed perturbed replication in SMC5-depleted cells, it is conceivable that the phosphorylation of p53 during prophase could be a consequence of late replication defects during G2 or the presence of under-replicated DNA, while the accumulation of p-p53 during G1 phase may be caused by the inheritance of unresolved DNA damage by G1 daughter cells.

Furthermore, we showed that the proliferation defect following AID-mediated depletion of SMC5 in mESCs was alleviated by inhibiting p53 or CHEK2, but not CHEK1. It was recently shown that depletion of SMC5/6 components in human cancer cells (HCT116) or hTERT immortalized non-cancer cells (RPE1) caused activation of CHEK2 and stabilization of p53 (*Venegas et al., 2020*). However, inactivation of *TP53* did not alleviate the cell-cycle arrest, cell death, and senescence observed following SMC5/6 degradation in either HCT116 or RPE1 cells (*Venegas et al., 2020*). This discrepancy may be attributed to the fact that our studies were performed in vivo in multipotent NPCs or in pluripotent mESC cultures, whereas HCT116 and RPE1 are tissue-specific cell lines.

## SMC5/6 is critical for completion of DNA replication

Although an earlier study using human RPE1 cells showed slower progression of DNA replication upon RNAi-mediated depletion of SMC6, our previous work demonstrated timely entry and exit from S phase in SMC5-deficient mESCs (*Gallego-Paez et al., 2014*; *Pryzhkova and Jordan, 2016*). However, our DNA fiber analyses indicate that replication forks encounter more perturbations upon SMC5 depletion. Perturbed replication fork progression may be partially counteracted by CHEK1-mediated activation of compensatory mechanisms, such as increased firing of dormant origins. Indeed, new origin firing was slightly elevated in SMC5-depleted mESCs. In our studies, apoptosis was shown to be primarily mediated by CHEK2. This does not rule out the possibility that CHEK1 could play a role, although no rescue of cell growth was observed with CHEK1 inhibition. Increased replication stress suggests that mESCs and NPCs likely proceed through anaphase with under-replicated DNA, potentially causing mitotic DNA damage (*Mankouri et al., 2013*; *Voutsinos et al., 2018*).

DNA fiber studies in budding yeast have shown disrupted replication fork progression in *Smc6* mutant cells under exogenous replication stress, but with no effects under unstressed conditions (*Bermúdez-López et al., 2010*). This suggests that SMC5/6 may be particularly important when replicative DNA damage levels are elevated. In contrast to the results presented here, a recent study using HCT116 cancer cells observed that the depletion of SMC5/6 complex components did not affect replication fork speed or CldU/IdU ratio (*Venegas et al., 2020*). However, mESCs and NPCs have higher levels of intrinsic replication stress due to their rapid rate of proliferation (*Ahuja et al., 2016*; *Arai et al., 2011*; *Ge et al., 2015*; *Waisman et al., 2019*). Indeed, the cell cycle duration of mESCs and NPCs is approximately twofold shorter than that of HCT116 cells (*Jensen et al., 2015*; *Waisman et al., 2019*). Another study reported that the Saos2 alternative lengthening of telomeres (ALT) cancer cell line requires SMC5/6 to promote telomere clustering and MiDAS at telomeres (*Min et al., 2017*). Although our findings in mESCs and NPCs demonstrate that SMC5/6 suppresses MiDAS, it is important to note that the ALT mechanism is a unique circumstance that requires distinct HR-mediated processes (*Cho et al., 2014*). SMC5/6 has previously been shown to be critical for telomere clustering, and NSMCE2-mediated SUMOylation of shelterin components is important for ALT (*Potts and Yu, 2007*).

The consequences of perturbed replication in SMC5-depleted NPCs and mESCs clearly manifest later in the cell cycle. This is evidenced by the increased formation of EdU+ and FANCD2+ foci during mitosis, particularly at late-replicating regions such as pericentromeric heterochromatin. Notably, SMC5/6 localizes to pericentromeric heterochromatin, as revealed by our previous work in mESCs, MEFs, and germ cells, suggesting that it may play a direct role at these sites (*Gómez et al., 2013*; *Hwang et al., 2018*; *Hwang et al., 2017*; *Pryzhkova and Jordan, 2016*; *Verver et al., 2013*). The relatively large percentage of EdU-FANCD2+ foci in both control and IAA-treated cells diverges from previous findings in U2OS cells, in which only one-third of FANCD2+ foci were EdU- (*Bhowmick et al., 2016*). This could be related to differences in the functions of FANCD2 in mESCs compared to human cancer cells. Interestingly, it has been shown that FANCD2 is particularly

important for promoting MiDAS in non-cancer cells (*Graber-Feesl et al., 2019*). It has been shown that FANCD2 is highly expressed in mESCs, though no comparison has been made with human cancer cells, and detailed studies of FANCD2 localization in mESCs have not been conducted (*Zhang et al., 2017*). Moreover, the absence of FANCD2 in a large percentage of EdU+ foci suggests that other non-FANCD2-mediated pathways could be at play during MiDAS. However, little is currently known about the factors involved in MiDAS in mESCs.

FA factors were recently shown to function upstream of SMC5/6 in the repair of exogenously induced inter-strand crosslinks (ICLs) (*Rossi et al., 2020*). Among other roles, FA factors are important for MiDAS and the resolution of replication termination regions, which resemble ICLs and are also genomic fragile sites (*Dewar and Walter, 2017*). . The increased prevalence of double-stalled converging forks in DNA fibers from SMC5-depleted cells likely reflects destabilization of these termination sites. These observations highlight an important role for SMC5/6 in fragile site replication. This is further reaffirmed by the fact that Smc5/6 is telomere-associated and involved in ribosomal DNA (rDNA) replication in yeast and human cells, with Smc5/6 deficiency leading to rDNA segregation errors during mitosis (*Moradi-Fard et al., 2016*; *Peng et al., 2018*; *Venegas et al., 2020*). Notably, SMC5 has also been identified at early replicating fragile sites in murine B lymphocytes via ChIP-seq studies (*Barlow et al., 2013*).

Although its precise role in replication fork stabilization is yet unknown, SMC5/6 is likely involved in the late steps of HR, as it functions downstream of FA factors and RAD51, of which we see an accumulation in *Smc5* cKO NPCs. RAD51 can mediate the formation of toxic recombination intermediates, and deletion of *RAD51* has been shown to rescue the lethality of *SMC6* mutants in fission and budding yeast (*Ampatzidou et al., 2006*; *Lehmann et al., 1995*; *Menolfi et al., 2015*). We previously showed that RAD51 foci are increased in *Smc5* cKO MEFs following hydroxyurea treatment (*Gaddipati et al., 2019*). Together, these observations suggest that SMC5/6 is necessary for the stabilization of stalled replication forks, rescue of collapsed replication forks, and repair of replication and recombination intermediates. The impairment of HR completion during replication leads to under-replicated DNA and unresolved replication intermediates. The presence of EdU+ foci on chromosomes of SMC5-depleted mitotic mESCs reflects an attempt to repair these intermediates. However, in the absence of SMC5, late stages of DNA repair cannot be completed, rendering these regions prone to chromatin missegregation. Under-replicated regions that persist in anaphase manifest as DNA bridges and can lead to DNA breaks in G1 daughter cells and subsequent CHEK2-mediated DDR (*Shaltiel et al., 2015*; *Voutsinos et al., 2018*). Collectively, these findings provide a molecular mechanism for the increased genome instability and chromosome segregation defects observed in SMC5/6-deficient cells. Future experiments will determine whether ablation of CHEK2 alleviates the accumulation of RAD51 or exacerbates this defect. Furthermore, it will be valuable to explore whether inhibition of HR can also alleviate the consequences of SMC5 depletion.

## Conclusions

We demonstrate that loss of SMC5/6 functions in the developing cortex leads to a reduction in cortex size, mislocalization of NPCs, reduced numbers of IPs and cortical layer neurons, and increased cell apoptosis. These developmental defects are attributed to SMC5/6 being required for proficient DNA replication, particularly at pericentromeric heterochromatin. SMC5/6 depletion causes NPCs to enter mitosis with regions of under-replicated DNA, which results in chromosome segregation defects. These defects stimulate a CHEK2-mediated DNA damage checkpoint response and result in elevated p53-mediated apoptosis (*Figure 7*).

## Materials and methods

### Animal use and care

All mice were bred at Johns Hopkins University (JHU, Baltimore, MD) in accordance with the National Institutes of Health and U.S. Department of Agriculture criteria and protocols for their care and use were approved by the Institutional Animal Care and Use Committees (IACUC) of JHU.

Mice of following strains were used: C57BL/6J (B6/J), stock number 000664 (Jackson Laboratory (JAX)); B6.129S2-*Emx1*tm1(cre)Krj/J (*Emx1-Cre*), stock number 005628 (JAX); B6.Cg-Tg(*Nes-Cre*)1Kln/J (*Nestin-Cre*), stock number 003771 (JAX); B6.129S2-*Trp53*tm1Tyj/J (*p53del* allele), stock number

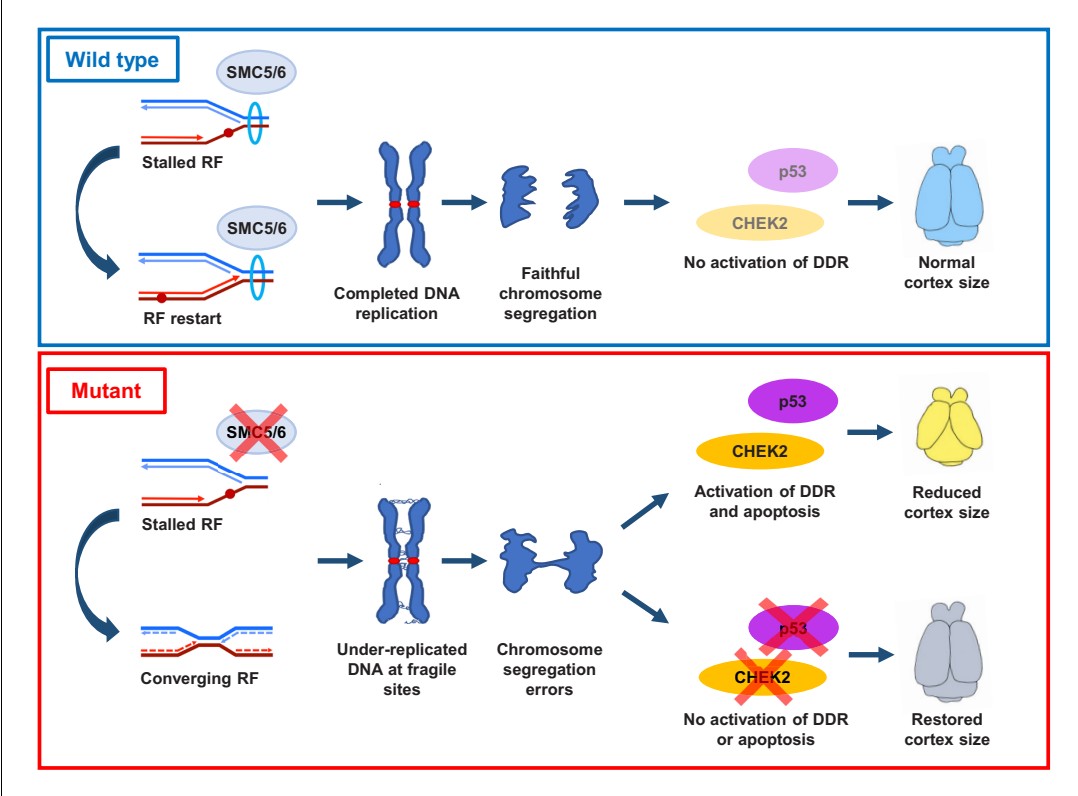

**Figure 7.** Proposed SMC5/6 complex functions in mammalian cortical development. SMC5/6 complex depletion in developing mouse cortex causes increased DNA replication stress at fragile sites, such as late-replicating heterochromatin. Unrepaired DNA and DNA intermediates persist into mitosis resulting in chromosome segregation defects. Acquired DNA damage triggers DNA damage response (DDR) and CHEK2- and p53-mediated apoptosis, resulting in reduced cortex size. Abrogation of p53 and CHEK2 functions alleviates SMC5 depletion phenotype. RF = replication fork; DDR = DNA damage response.

002101 (JAX); *Chek2^{tm1b(EUCOMM)Hmgu}* (*Chek2^{del}* allele), stock number 047089-UCD (University California Davis); B6;129-*Trp53bp1^{tm1Jc}*/J (*Trp53bp1^{del}* allele), stock number 006495 (JAX). Mice harboring *Smc5* with a floxed exon 4 (designated *Smc5^{flox}*) and deleted exon 4 (designated *Smc5^{del}*) have been previously described (**Hwang et al., 2018**; **Hwang et al., 2017**; **Pryzhkova and Jordan, 2016**).

These mice were bred to obtain the following genotypes: *Smc5^{flox/flox}*, *Emx1-Cre tg/0* (hemizygous transgene); Smc5^{flox/flox}, *Nestin-Cre tg/0*; *Smc5^{flox/flox}*, *Trp53^{del/del}*, *Emx1-Cre tg/0*; *Smc5^{flox/flox}*, *Trp53bp1^{del/del}*, *Emx1-Cre tg/0*; *Smc5^{flox/flox}*, *Chk2^{del/del}*, *Emx1-Cre tg/0*. Mice from both genders were included in the study for all ages. Genotypes not resulting in homozygous gene knockout were used as controls.

## Mouse genotyping

PCR genotyping was performed using AccuStart II PCR SuperMix (Quanta BioSciences).

Primers used are described in **Supplementary file 1**. PCR reaction conditions were as follows: initial denaturation at 94°C for 2 min (denaturation at 94°C for 20 s, annealing at 58°C for 30 s, amplification at 72°C for 30 s) × 34 cycles, and final extension at 72°C for 10 min. For *Trp53bp1*, PCR conditions were as follows: initial denaturation at 94°C for 2 min (denaturation at 94°C for 20 s, annealing at 65°C for 15 s with 0.5°C per cycle decrease) × 10 cycles; (denaturation at 94°C for 15 s, annealing at 60°C for 15 s, amplification at 72°C for 10 s) × 28 cycles, and final extension at 72°C for 2 min.

## Behavioral studies

### Adhesive patch test
The test was performed as previously described (*Fleming et al., 2013*). A half of circular paper adhesive (9.5 mm diameter) was placed on right rare paw and time to contact was recorded. Each mouse was given up to 60 s to remove the paper adhesive. The mice were alternated to have three trials each for every age time point assessed.

### Inverted screen test
The test was performed as previously described (*Grady et al., 2006*). A mouse was placed in the middle of wire mesh grid (16 squares per 10 cm), and screen was inverted to 180°. A mouse was timed for how long it remained upside down on the screen. Two trials were administered for each animal with 20–30 min interval between trials. Means were calculated across the trials for each mouse.

### Cylinder test
The test was performed as previously described (*Fleming et al., 2013*). A mouse was placed into glass cylinder (15 cm diameter) and video recorded for 3 min. Videos were analyzed, and the number of rears with forelimbs touching the glass wall or not touching it was scored.

## Western blot analysis
Cell lysates were prepared in RIPA buffer (Santa Cruz Biotechnology) supplemented with protease inhibitor cocktail (Roche). Equal amounts of proteins were fractionated by SDS-PAGE and transferred to PVDF membrane (Bio-Rad). Primary and secondary antibody information is provided in *Supplementary file 2*. We used horseradish peroxidase (HRP)-conjugated goat anti-mouse-IgG and anti-rabbit-IgG secondary antibodies (Invitrogen). Signal was detected using Clarity Western ECL Substrate (Bio-Rad) and imaged using Syngene XR5 system.

## Immunohistochemistry
Mouse brains and heads were collected and fixed in 10% of formalin solution (Sigma-Aldrich): E13.5 brains were fixed for 1.5 hr, E16.5 brains for 3 hr, P0/1 brains for 6 hr, P0/1 heads and adult P55-58 brains overnight at 4°C. Following fixation, tissues were transferred to a 20% sucrose solution in PBS and left overnight at 4°C. Tissues were then transferred into O.C.T. compound (Scigen) and frozen at −80°C. Coronal or sagittal sectioning of frozen O.C.T. blocks was performed at 16 μm thickness using the Cryo3 (Sakura Tissue-Tek). All sections were mounted on TruBOND 380 (Matsunami) or Selectfrost Adhesion (Fisher) microscope slides. Sections were kept at −80°C until further processing.

Immunohistochemistry was carried out with or without antigen retrieval (see *Supplementary file 2*). Antigen retrieval was performed in sodium citrate buffer (10 mM sodium citrate, 0.05% Tween-20, pH 6.0) in a 70°C water bath for 20 min. All sections (regardless of antigen retrieval status) were incubated with permeabilizing and blocking solution (0.25% Triton X-100, 10% horse serum in PBS) for 1 hr at room temperature in a humidified slide box. Sections were incubated with primary antibodies diluted in blocking solution (10% horse serum in PBS) overnight at 4°C in a humidified slide box. Sections were washed three times with TBS-T rinse buffer (20 mM Tris-HCl, pH 7.4, 150 mM NaCl, 0.05% Tween-20) and incubated with secondary antibodies for 1 hr at room temperature in a humidified slide box. After washing with TBS-T buffer sections were mounted using Vectashield with DAPI (Vector Laboratories). Antibody information is provided in *Supplementary file 2*.

## TUNEL assay
Identification of apoptotic cells in brain cryosections was performed using In Situ BrdU-Red DNA fragmentation (TUNEL) assay kit (Abcam). Sections were mounted using Vectashield with DAPI (Vector Laboratories).

## CldU labeling
Pregnant females at 16.5-day post-coitum were injected intraperitoneally with 50 μg/g of body weight CldU (Sigma). Mice were sacrificed 4 hr after injection, and embryonic brains were collected

and processed for cryosectioning as described above. For CldU immunostaining, frozen sections were subject to antigen retrieval as described above, followed by permeabilization with 0.25% Triton X-100, 10% horse serum in PBS for 30 min at room temperature, 30 min 2 M HCl treatment at 37°C, then blocking and incubation with primary and secondary antibodies for 2 hr at room temperature each. After washing with TBS-T buffer sections were mounted using Vectashield with DAPI (Vector Laboratories). Antibody information is provided in *Supplementary file 2*.

## Mouse cortical cell immunocytochemistry

E16.5 cortices were dissociated with 0.05% trypsin-EDTA (Gibco) into single cells and placed into cell culture medium in the presence of 10 µM STLC (Tocris) for 1 hr to enrich mitotic cells and then collected for analysis. Chromosome spread preparation from E16.5 primary cortical cell cultures was performed as described previously (*Pryzhkova and Jordan, 2016*). Immunocytochemistry was performed as described previously (*Pryzhkova et al., 2014*). Antibodies used are listed in *Supplementary file 2*. Samples were mounted using Vectashield with DAPI (Vector Laboratories).

## mESC culture and analysis

B6 mESCs used in this study were established and maintained in 2i/LIF medium as described in *Pryzhkova and Jordan, 2016*; *Pryzhkova et al., 2020*. mESCs were verified to be negative for mycoplasma using the PCR Mycoplasma Test Kit I/C (PromoCell). For cell growth analysis mESCs were cultured in the presence of 10 µM CHEK2 inhibitor II (Cayman), 3 µM CHEK1 inhibitor LY2603618 (Cayman), or p53 inhibitor cyclic pifithrin-alpha hydrobromide (Cayman), with or without 100 µM IAA for 48 hr. Drugs were added 18–20 hr after passaging. mESCs were counted at the time of passaging and after 48 hr of growth in the presence of drugs.

## DNA fiber assay

For DNA fiber assay, mESCs were treated with CHEK2 inhibitor, CHEK2 inhibitor and IAA, p53 inhibitor, or p53 inhibitor and IAA 16–18 hr after passaging. After 12 hr of treatment, mESCs were incubated in culture with 30 µM CldU (Sigma) for 20 min, washed twice with PBS, and incubated with 250 µM iododeoxyuridine (IdU) (Sigma) for 20 min. Labeled mESCs were resuspended in PBS at $2 \times 10^5$ cells/ml. DNA fiber spreading and immunostaining were performed as previously described (*Huang et al., 2013*). Primary antibodies used were rat anti-BrdU (CldU) (Abcam) and mouse anti-BrdU (IdU) (Becton Dickinson). Secondary antibodies used were Alexa Fluor anti-rat 568 and Alexa Fluor anti-mouse 488. Antibody information is provided in *Supplementary file 2*.

## mESC immunocytochemistry

For MiDAS assessment, mESCs were treated with IAA 16–18 hr after passaging. After 18 hr of IAA treatment, mESCs were incubated with 8 µM RO-3306 (Sigma) for 6 hr, washed twice in PBS, and cultured in the presence of 10 µM STLC (Tocris) and 10 µM ethynyl-deoxyuridine (EdU) (Sigma) for 1 hr. mESCs were washed and collected. Chromosome spread preparation was performed as described previously (*Pryzhkova and Jordan, 2016*). For EdU detection, chromosome spreads were washed three times in PBS and incubated with 'click' reaction cocktail containing 0.1 M Tris (pH 8.5), 10 µM cyanine 5-azide (Lumiprobe), 1 mM $CuSO_4$, and 0.1 M L-ascorbic acid (Sigma) added last. All reaction components were dissolved in 20% dimethylsulfoxide (DMSO) (Sigma) in PBS. Chromosome spreads were incubated with the reaction cocktail for 20 min and washed in PBS with 0.5% Triton three times for 10 min each. Immunocytochemistry was performed as described previously (*Pryzhkova et al., 2014*). Antibodies used are listed in *Supplementary file 2*. Samples were mounted using Vectashield with DAPI (Vector Laboratories).

## Microscopy

Images were captured using a Zeiss Cell Observer Z1 fluorescence microscope linked to an ORCA-Flash 4.0 CMOS camera (Hamamatsu), or Zeiss AxioImager A2 linked to AxioCam ERc 5 s camera (Zeiss), or Keyence BZ-X800 fluorescence microscope. Images were analyzed and processed using ZEN 2012 blue edition imaging software (Zeiss) or with BZ-X800 Viewer and Analyzer software (Keyence). Photoshop (Adobe) was used to prepare figure images.

## Image data quantification

### Brain section analysis

The measurements of cortical area were made using Fiji (ImageJ) (*Schneider et al., 2012*). To determine the ratio of cortical layers to cortex thickness, we measured the thickness of the cortical layer being analyzed and the thickness of the neural cortex (from the apical ventricular surface to the basal surface). This was performed at 90° to the ventricular surface using the angle tool in Fiji (ImageJ). The thickness of hematoxylin and eosin-stained cortical sections was measured at 90° to the ventricular surface using the angle tool in Fiji (ImageJ). For counts of cells expressing specific marker, cells were counted within the defined area. Similar regions of the cortex were compared between representative genotypes.

Anaphase-stage mitotic cells were classified in the VZ area. The cleavage plane orientation of anaphase cells in the VZ was defined by the angle between the cleavage plane and the ventricular surface. For cleavage plane orientation, mitotic cells were scored as vertical (60–90°), oblique (30–60°), and horizontal (0–30°). Cells were quantified using Fiji (ImageJ). Detailed information about cortical section analysis is provided in *Supplementary file 3*.

### Chromosome spreads analysis

For measurements of H4K20 signal width on cortical cell spreads, a line was drawn across the widest part of the pericentromeric H4K20 signal, and an additional line was drawn perpendicular to this line. The length of both lines was measured and averaged. This was performed for three pericentromeric H4K20 signals per cell, and the average of the three measurements was calculated to determine the average width of the pericentromeric H4K20 signal in each cell. For H4K20 signal intensity, the H4K20 channel image was converted to grayscale, the pericentromeric H4K20 signal was outlined, and the average pixel intensity was measured within the outlined area. This was performed for three pericentromeric H4K20 signals per cell, and the average of the three measurements was calculated to determine the average intensity of the pericentromeric H4K20 signal in each cell. The average intensity was normalized to the background intensity of the image, which was determined by measuring the average pixel intensity of the grayscale image in the area where there were no chromosome spreads. Image data quantification was performed using Fiji (ImageJ).

For FANCD2, EdU, and CEN assessments in cortical cells and mESCs, foci were counted using the multi-point tool in Fiji (ImageJ).

### DNA fiber analysis

For measurements of DNA fiber length, a line was drawn along each dual-labeled DNA fiber. The length of the line was measured. For assessment of DNA fiber CldU/IdU ratio, a line was drawn along the CldU-labeled section and the IdU-labeled section of each dual-labeled fiber, and the length of the lines measured. The ratio of CldU length to IdU length was calculated. Image data quantification was performed using Fiji (ImageJ).

## Statistical analysis

Statistical analyses were performed using GraphPad Prism V5/8 software and RStudio. For anaphase counts, cleavage plane orientation, and NPC percentage in VZ and outside of VZ statistical significance was assessed using a chi-squared test with Yates' correction for continuity. For quantification of the western blot intensity in *Figure 2A* and *Figure 2—figure supplement 1A* the significance was assessed using an unpaired two-tailed Student's t-test. For all other assessments, a non-parametric unpaired two-tailed Mann–Whitney U-test was used. p-values of less than 0.05 were considered significant. All data represent the means ± S.E.M. unless noted otherwise. *$p < 0.05$; **$p < 0.01$; ***$p < 0.001$, ****$p < 0.0001$ and ns (not significant) indicates $> 0.05$. Individual p-values for all graphs presented in each figure are available in *Supplementary file 3*.

## Acknowledgements

We thank Ewelina Bolcun-Filas for the *Chek2* KO mouse, Thao Phan and Andrew Holland for discussion and reagents, and Jing Zhang and Michael Seidman for guidance with DNA fiber assay.

## Additional information

### Funding

| Funder | Grant reference number | Author |
|---|---|---|
| National Institute of General Medical Sciences | R01GM11755 | Philip W Jordan |
| National Institutes of Health | R21OD023720 | Philip W Jordan |
| National Institute of Neurological Disorders and Stroke | R03NS106486 | Philip W Jordan |
| Johns Hopkins University | Catalyst Award | Philip W Jordan |
| National Cancer Institute | T32CA009110 | Michelle J Xu |

The funders had no role in study design, data collection and interpretation, or the decision to submit the work for publication.

### Author contributions

Alisa Atkins, Maggie Li, Data curation, Formal analysis, Validation, Investigation, Visualization, Methodology, Writing - original draft, Writing - review and editing; Michelle J Xu, Data curation, Formal analysis, Validation, Investigation, Visualization, Methodology, Writing - original draft; Nathaniel P Rogers, Data curation, Formal analysis, Validation, Investigation, Visualization, Methodology; Marina V Pryzhkova, Data curation, Formal analysis, Supervision, Validation, Investigation, Visualization, Methodology, Writing - original draft, Project administration, Writing - review and editing; Philip W Jordan, Conceptualization, Resources, Data curation, Formal analysis, Supervision, Funding acquisition, Validation, Investigation, Visualization, Methodology, Writing - original draft, Project administration, Writing - review and editing

### Author ORCIDs

Maggie Li ![ORCID] http://orcid.org/0000-0003-1047-1554
Nathaniel P Rogers ![ORCID] http://orcid.org/0000-0002-0411-5249
Marina V Pryzhkova ![ORCID] https://orcid.org/0000-0002-3462-5768
Philip W Jordan ![ORCID] https://orcid.org/0000-0003-4890-2647

### Ethics

Animal experimentation: All mice were bred at Johns Hopkins University (JHU, Baltimore, MD) in accordance with the National Institutes of Health and U.S. Department of Agriculture criteria and protocols for their care and use were approved by the Institutional Animal Care and Use Committees (IACUC) of JHU (Protocol number = MO19H08).

### Decision letter and Author response

Decision letter https://doi.org/10.7554/eLife.61171.sa1
Author response https://doi.org/10.7554/eLife.61171.sa2

## Additional files

### Supplementary files

- Supplementary file 1. Primers used in this study.
- Supplementary file 2. Antibodies used in this study.
- Supplementary file 3. Statistics and p-values.
- Transparent reporting form

### Data availability

All data generated or analysed during this study are included in the manuscript and supporting files.

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
