## [Decision Letter]

Thank you for submitting your article "SMC5/6 is required for replication fork stability and faithful chromosome segregation during neurogenesis" for consideration by *eLife*. Your article has been reviewed by three peer reviewers, and the evaluation has been overseen by Kevin Struhl as the Senior and Reviewing Editor. The reviewers have opted to remain anonymous.

The reviewers have discussed the reviews with one another and the Reviewing Editor has drafted this decision to help you prepare a revised submission.

The reviewers have indicated a number of changes in the text to soften the conclusions, and these are required. In addition, the following experiments are strongly suggested to be done:

1) Show when p53/*chek2* are activated upon *Smc5* loss in mESC cells.

2) Show DNA fiber results for IAA treatment alone samples as an important control. The revised manuscript should be submitted with point-by-point answers.

Reviewer #1:

The authors developed two *Smc5* cKO mice lines to conditionally deplete *Smc5*/*6* in central cortex tissues using tissue specific Cre constructs. They verified the reduction of *Smc5* and *6* protein levels in E16.5 embryos and to a less degree in E13.5 embryos. The resultant mice exhibited increased apoptosis, mitotic DNA abnormalities, increase of Rad51 levels and oblique division axis in NPCs, and consequently reduced brain weight and cortex size in newborn and adult mice, as well as showed sensorimotor function defects. Deleting *Chek2* or p53, but not 53BP1, significantly improved various *Smc5* cKO mice defects regarding brain weight, cortex area, and cortical thickness, apoptosis, and DNA fragmentation, suggesting that the DSB-induced DDR mediated by the ATM-*Chek2*-p53 axis is activated in *Smc5* KO mice and responsible for many defects in these cells. p53/Check2 KO did not improve the ectopic localization of apical NPC cells marked by *SOX2*/PAX6 seen in *Smc5* KO mice. Both *Smc5* KO and combined KO with p53/*Chek2* increases total *SOX2*/PAX6+ NPC cells, which was interpreted as compensator effects. In contrast to apical NPCs, decrease in several other types of neuron in *Smc5* KO mice was rescued by p53/*Chek2* KO. How to explain this contrast effect of p53/*Chek2* KO in different type of cells should was not clear.

Staining assays suggests that *Smc5* KO NPCs tend to accumulate in G1, an effect reversible by *Chek2* KO. Image in Figure 4H suggests anaphase bridge with H3P marking ends of the DNA bridges. It is unclear how prominent this phenotype is. Levels of FACND2 foci, including those co-localizing with pericentrimeric regions, increased about 6 to 8-fold in the chromosome spread of *Smc5* KO cells. In addition, level of H4K20me3 at pericentromeric regions increased 2X in *Smc5* KO cells.

The rest of the paper describes studies in mESCs cells by acute *Smc5* depletion upon IAA addition. *Smc5* depletion retards cell growth, which is reversed by cotreatment by *Chek2i* or p53i, while later addition of *Chek2i* had no effect. DNA fiber results suggest that compared to *Chek2i* (or p53i) treated cells, IAA+*Chek2i* (or IAA+p53i) treatment reduced ongoing DNA synthesis but appears to increase stalled and converging replication. Experiments should be shown to address whether these effects can be seen upon IAA treatment compared with untreated conditions without *Chek2i*. The fact that *Chek2i* and IAA+ *Chek2i* treatment are comparable for cell growth but not for replication is puzzling. Additional data are provided to suggest increase MiDAS in +IAA conditions. Taken together, the authors conclude that *Smc5*/*6* helps to complete replication before mitosis and thus is critical in highly proliferative stem cells during development.

This work generates the first neuron specific *Smc5* KO mice and demonstrates that *Smc5* loss leads to multiple brain development defects, which can provide a nice explanation for NSMCE2 human patient microcephaly phenotype. This reviewer feels that the developmental data quality present here is quite high, though an expert in this area should be consulted. Another important conclusion of the paper is that many NPC defects associated with *Smc5* KO are caused by *Chek2*/p53 DDR mediated cell death. Phenotype of SMC5 KO mESC cells are largely expected from similar studies and can provide an explanation for *Smc5* NPC KO mice phenotype. In particular, this work shows that *Smc5* KO in mESCs and in mice can respond differently to p53/check2 loss than in cancer cell lines as recently reported by Vegengas et al., 2020. With this said, interpretations of mESC data here needs to be more carefully vetted, as most data are not causative. For example, increased levels of MiDAS may not necessarily be caused by S phase defects. Could *Smc5* depletion in mitosis or G2 phase be sufficient to cause increased MiDAS? Also, it is unclear whether increased Rad51 levels is one of the causes for replication defects seen in *Smc5* KO cells. Does p53 or *chek2* KO rescue the increased Rad51 levels of *Smc5* KO mice? What types of events initially trigger the p53/*chek2* signaling pathway in *Smc5*/*6* KO cells, are these replication-born DSBs? Standard analysis to directly measure p53/*Chek2* activation can help to pinpoint when cells evoke this pathway, thus help to formulate a model to tie various defects reported here.

Reviewer #2:

This article from the Jordan laboratory describes an extensive set of analyses on a new mouse model where the *Smc5* gene was conditionally inactivated. These analyses were complemented by studies on *Smc5* degron cells. There have been several previous reports of the effects of impairing *Smc5*/*6* functions in mouse or human cells, but the current study extends those analysis by focusing on effects in vivo and, more specifically, in the developing neocortex. Overall, I really liked the article and the clear manner in which the analyses were described. I think that it will be an important addition to the many times confusing array of studies on *Smc5*/*6* function. There were, however, some concerns with the cellular analyses in particular.

1) In several places it is discussed that pericentromeric heterochromatin is a common fragile site. My understanding is that, at least in human cells, this heterochromatic region very rarely, if at all, shows gaps or breaks on mitotic chromosomes. Further, the marker of fragile sites used here, FANCD2, also does not generally localize to pericentromeric regions. I found the discussion around this point quite contrived as if it was intended to link the fact that *Smc5*/*6* localizes to pericentromeric regions with some well recognized manifestation of replication stress like fragility (and MiDAS that they used) when there isn't such a clear association. The authors cite Bhowmick et al. as the reference for EdU foci (MiDAS) and FANCD2 being at centromeric heterochromatin (subsection “SMC5/6 is critical for completion of DNA replication”), but I could find no reference to pericentromeric regions in that paper.

2) Related to 1) it would be important to show that real gaps/breaks and MiDAS foci co-localize with at centromeric heterochromatin regions – and that they are distinct from the telomeres that are adjacent in mouse chromosomes. Telomeres are hotspots for fragility and MiDAS. On a related note, the red and pink staining in Figure 6 is really hard to distinguish – couldn't the authors pseudo-color the foci differently? It is important to see where the telomere is on the acrocentric chromosome end. The impression I get is that many of the FANCD2 and EdU foci are at telomeres. These images do not look very crisp either and the resolution could be improved by collecting better confocal images I would suggest.

3) Related to 2), what is the percentage of EdU foci in mitosis that co-localize with FANCD2? Unlike what has been shown in previous studies on human cancer cell lines, many of the EdU foci shown here don't seem to co-localize with FANCD2. This should be quantified and discussed in the context of the current literature.

4) The fiber analyses should be repeated with a new control where CHEK2 inhibition is not employed in all of the samples, including the so-called negative control. As things stand, it is hard to compare the current analysis with previous analyses done with controls lacking any added compounds.

Reviewer #3:

This is a nicely conducted analysis of conditional SMC5 deletion in mouse, with well executed and controlled experiments and robust data. The work does not uncover any major surprises or unexpected discoveries in terms of SMC5 function/cellular phenotypes, but does consolidate and demonstrate previously established and anticipated roles in an in vivo setting. The mice developed here have been carefully characterised and will be of use as disease models by this and other laboratories.

The primary finding is that SMC5/6 is needed for tolerance to replication stress and replication fork stability/restart, and consequently for normal neurogenesis. The authors also show convincingly that defects in SMC5 function elicit cellular and organismal pathology via the CHEK2/p53 pathway. I have no major criticisms. The work will be of broad interest to the chromosome structure and maintenance research fields, and particularly so to investigators interested in cohesion/condensin function/dysfunction.

---

## [Author Response]

The reviewers have indicated a number of changes in the text to soften the conclusions, and these are required. In addition, the following experiments are strongly suggested to be done:1) Show when p53/chek2 are activated upon Smc5 loss in mESC cells.

To investigate p53 and *Chek2* activation in *Smc5*-depleted mouse embryonic stem cells (mESCs) we performed following analyses:

(1a) western blot analyses of mESCs treated with IAA at multiple timepoints demonstrated that acetyl-p53 (K379) and phospho-p53 (S389) are upregulated after 2 hours and 48 hours of auxin treatment, respectively. These data are added to Figure 5C, D and discussed in the Results section. We were unable to obtain conclusive results with phospho-CHEK2 antibodies due to their non-specificity and non-reactivity in mESCs (see details in the response to reviewer 1).

(1b) using immunofluorescence analysis, we also assessed the cell cycle distribution of phospho-p53+ mESCs using proliferative cell markers Ki67 and H3P (H3 serine 10 phosphorylation). Ki67 staining demonstrated that close to 100% of cells were proliferative. H3P staining in mESCs has been previously characterized to show that H3P foci are present on DNA during S-phase adjacent to DAPI-dense regions that become more prominent by G2, colocalizing with the pericentromeric heterochromatin (Mallm and Rippe, 2015; Pryzhkova and Jordan, 2016). By M-phase H3P localizes throughout the condensed chromatin. Therefore, based on H3P expression, we defined G1 (H3P-), S/G2 and mitotic stages (H3P+) of cell cycle in mESCs and quantified the distribution of phospho-p53 positive cells. Details are provided in the Results section. We observed an increase in the percentage of phospho-p53+ cells in both H3P+ and H3P- populations after 48 hours of IAA treatment. The abundance of phospho-p53+ cells in prophase and G1 populations compared to S/G2 cells suggests that p53 (S389) phosphorylation is most likely to occur in prophase or G1 phases upon SMC5 depletion. Specifically, 38.2% of G1 phase cells, 26.2% of S/G2 phase cells and 40.8% of prophase cells were positive for p-p53 in the IAA-treated group compared to 11.9%, 11.8% and 7.0% in control, respectively. We also observed a 1.5-fold increase in the percentage of total G1 phase cells, a 1.5-fold decrease in the percentage of total S/G2 phase cells, and a 2.1-fold increase in the percentage of total prophase cells in the IAA-treated group compared to control. These data are shown in Figure 5—figure supplement 1D-F and discussed in the Results section.

Unfortunately, we did not see a discernible difference in acetyl-p53 intensity in mESCs treated with various durations of IAA, as a high degree of background staining was present in both control and treated cells. Also, we were unable to assess activation of CHEK2 in mESCs due to non-specificity and non-reactivity of the phospho-CHEK2 antibodies we tested (see details in the response to reviewer 1).

(1c) As an additional experiment, to validate the reactivity of phospho-CHEK2, phospho-p53 and acetyl-p53 antibodies used for mESC analyses and complement our data, we performed immunofluorescence assessments of mouse E16.5 cortical sections. We found that acetyl-p53 was upregulated in the progenitor zone of *Smc5* cKO cortical sections compared to control. We have added these data to Figure 2—figure supplement 1B and discussed it in the Results section. We did not observe any staining with phospho-CHEK2 and phospho-p53 antibody in *Smc5* cKO and control cortical sections due to non-reactivity.

In summary, we show the upregulation of acetyl-p53 in *Smc5* cKO cortices and of acetyl-p53 and phospho-p53 in SMC5-depleted mESCs. We demonstrate that acetylation of p53 is an early response to SMC5 depletion in mESCs. In addition, we show that p53 (S389) phosphorylation occurs significantly later than p53 acetylation and is most frequently present in G1 phase and prophase cells. This suggests that aberrancies caused by SMC5 depletion (for example, late replication defects during G2) lead to a DNA damage response during prophase and potentially, the inheritance of DNA damage by G1 daughter cells.

2) Show DNA fiber results for IAA treatment alone samples as an important control. The revised manuscript should be submitted with point-by-point answers.

We performed DNA fiber experiments in the presence of IAA alone (without CHEK2 inhibitor). We have included these results in Figure 5H and Figure 5—figure supplement 1H, I and discussed it in the Results section. In brief, we show that DNA fibers from IAA-treated mESCs exhibit increased frequency of converging forks and stalled forks, and reduced incidence of ongoing forks, which is similar to the results with CHEK2 inhibition. IAA-treated mESCs also have slightly shorter DNA fiber track length and increased CldU/IdU length ratio, similar to the results with CHEK2 inhibition.

Reviewer #1:The authors developed two Smc5 cKO mice lines to conditionally deplete Smc5/6 in central cortex tissues using tissue specific Cre constructs. They verified the reduction of Smc5 and 6 protein levels in E16.5 embryos and to a less degree in E13.5 embryos. The resultant mice exhibited increased apoptosis, mitotic DNA abnormalities, increase of Rad51 levels and oblique division axis in NPCs, and consequently reduced brain weight and cortex size in newborn and adult mice, as well as showed sensorimotor function defects. Deleting Chek2 or p53, but not 53BP1, significantly improved various Smc5 cKO mice defects regarding brain weight, cortex area, and cortical thickness, apoptosis, and DNA fragmentation, suggesting that the DSB-induced DDR mediated by the ATM-Chek2-p53 axis is activated in Smc5 KO mice and responsible for many defects in these cells. p53/Check2 KO did not improve the ectopic localization of apical NPC cells marked by SOX2/PAX6 seen in Smc5 KO mice. Both Smc5 KO and combined KO with p53/Chek2 increases total SOX2/PAX6+ NPC cells, which was interpreted as compensator effects. In contrast to apical NPCs, decrease in several other types of neuron in Smc5 KO mice was rescued by p53/Chek2 KO. How to explain this contrast effect of p53/Chek2 KO in different type of cells should was not clear.

Firstly, we thank you for your time and effort in reviewing our manuscript. This is a very interesting question for further investigation. We speculate that *Smc5* depletion also causes perturbations in neuronal differentiation associated with increased mitotic defects and apoptosis (subsections “SMC5 loss causes abnormal cortical development” and “SMC5 depletion leads to increased mitotic DNA synthesis”). Neuronal differentiation in *Smc5* cKO is not completely abrogated, but corticogenesis is affected by apoptosis. In case of p53/*Chek2* KO rescue phenotypes, apoptosis is alleviated allowing for production of larger numbers of layer-specific neurons. However, all other affected developmental processes in rescue mice are not restored. This phenomenon might be difficult to investigate using mouse models, but possibly cell culture-based approaches can provide further information.

Staining assays suggests that Smc5 KO NPCs tend to accumulate in G1, an effect reversible by Chek2 KO. Image in Figure 4H suggests anaphase bridge with H3P marking ends of the DNA bridges. It is unclear how prominent this phenotype is.

H3P is always observed incorporated in the chromatin bridges in anaphase cells (see additional images included). However, detecting ultrafine DNA bridges on histological sections is technically challenging. Even a monolayer cell culture requires specific techniques to detect ultrafine DNA bridges successfully (Bizard et al., Genome Instability: Methods and Protocols, 2018). We attempted to perform immunostaining with anti-RAD51, PICH, FANCD2, RPA2 and BLM antibodies, albeit, unsuccessfully. The image shown in Figure 4H is a representative of cortical anaphase cell with very likely broken segments of chromosomes bound by fine DNA bridge, formed by under-replicated DNA, which is not detectable due to the scarce amount of chromatin. We incorporated these additional images in the manuscript to demonstrate the existence of these events. Nevertheless, improved techniques are required to be able to quantify defects such as these in an in vivo setting.

We included additional anaphase images into Figure 4—figure supplement 1G and the following text into the Results section: “The detection of ultrafine DNA bridges formed by under-replicated DNA on histological sections is technically challenging, and our data require further support. Thus, next we focused on the analysis of chromosome spreads prepared from dissociated E16.5 cortices.”

Levels of FACND2 foci, including those co-localizing with pericentrimeric regions, increased about 6 to 8-fold in the chromosome spread of Smc5 KO cells. In addition, level of H4K20me3 at pericentromeric regions increased 2X in Smc5 KO cells.The rest of the paper describes studies in mESCs cells by acute Smc5 depletion upon IAA addition. Smc5 depletion retards cell growth, which is reversed by cotreatment by Chek2i or p53i, while later addition of Chek2i had no effect. DNA fiber results suggest that compared to Chek2i (or p53i) treated cells, IAA+Chek2i (or IAA+p53i) treatment reduced ongoing DNA synthesis but appears to increase stalled and converging replication. Experiments should be shown to address whether these effects can be seen upon IAA treatment compared with untreated conditions without Chek2i.

We performed additional DNA fiber experiments in the absence or presence of IAA alone (without CHEK2 inhibitor). We have included these results in Figure 5H and Figure 5—figure supplement 1H, I and discussed it in the Results section. In brief, we show that DNA fibers from IAA-treated mESCs exhibit increased frequency of converging forks and stalled forks, and reduced incidence of ongoing forks, which is similar to the results with CHEK2 inhibition. IAA-treated mESCs also have slightly shorter DNA fiber track length and increased CldU/IdU length ratio, similar to the results with CHEK2 inhibition.

The fact that Chek2i and IAA+ Chek2i treatment are comparable for cell growth but not for replication is puzzling.

We attribute this to the fact that CHEK2 inactivation may prevent cell death, despite perturbations that would normally induce CHEK2 activation. In other words, cells may bypass apoptosis, but still harbor defects.

Additional data are provided to suggest increase MiDAS in +IAA conditions. Taken together, the authors conclude that Smc5/6 helps to complete replication before mitosis and thus is critical in highly proliferative stem cells durning development.This work generates the first neuron specific Smc5 KO mice and demonstrates that Smc5 loss leads to multiple brain development defects, which can provide a nice explanation for NSMCE2 human patient microcephaly phenotype. This reviewer feels that the developmental data quality present here is quite high, though an expert in this area should be consulted.

We received continual critique from Dr. Miho Iijima (Ph.D., Professor in Cell Biology, Johns Hopkins School of Medicine). Dr. Iijima together with Dr. Hiromi Sesaki (Ph.D., Professor in Cell Biology, Johns Hopkins School of Medicine) who specialize in studying neurodevelopment and neurodegenerative disorders. Furthermore, we communicated our research with our collaborators Dr. Mustafa Khokha (M.D., Professor of Pediatric Critical Care Medicine and Genetics, Yale School of Medicine), Dr. Laura Ment (M.D., Professor of Pediatrics Neurology) for additional feedback prior to submission.

Another important conclusion of the paper is that many NPC defects associated with Smc5 KO are caused by Chek2/p53 DDR mediated cell death. Phenotype of SMC5 KO mESC cells are largely expected from similar studies and can provide an explanation for Smc5 NPC KO mice phenotype. In particular, this work shows that Smc5 KO in mESCs and in mice can respond differently to p53/check2 loss than in cancer cell lines as recently reported by Vegengas et al., 2020. With this said, interpretations of mESC data here needs to be more carefully vetted, as most data are not causative. For example, increased levels of MiDAS may not necessarily be caused by S phase defects. Could Smc5 depletion in mitosis or G2 phase be sufficient to cause increased MiDAS?

We have edited the manuscript text to ensure we remove claims that may be construed that the phenotype observed in SMC5-depleted mESCs is specifically due to defects acquired in S phase.

Also, it is unclear whether increased Rad51 levels is one of the causes for replication defects seen in Smc5 KO cells.

We have not assessed the role of RAD51 in promoting or counteracting replication stress in *Smc5* cKO cells. However, RAD51 can mediate the formation of toxic recombination intermediates, and deletion of *RAD51* has been shown to rescue lethality in fission and budding yeast with *SMC6* mutations (Lehmann et al., 1995; Menolfi et al., 2015). We previously showed that RAD51 foci are increased in *Smc5* cKO MEFs following hydroxyurea treatment (Gaddipati, Pryzhkova, and Jordan, 2019). We have referenced these investigations in our Discussion and promote it as a promising avenue for further investigation.

Does p53 or chek2 KO rescue the increased Rad51 levels of Smc5 KO mice?

This is a question we agree warrants further investigation. However, it will require additional mice breeding and collection of embryos from multiple litters, as double KO mice only constitute a small proportion of embryos harvested. Under current COVID-19 restrictions it is very time-consuming task and would require at least half a year to answer this question. We will definitely be pursuing this line of questioning in the future and we reference this within our Discussion.

What types of events initially trigger the p53/chek2 signaling pathway in Smc5/6 KO cells, are these replication-born DSBs? Standard analysis to directly measure p53/Chek2 activation can help to pinpoint when cells evoke this pathway, thus help to formulate a model to tie various defects reported here.

First, we used western blot analysis to assess the acetylation and phosphorylation of p53 in mESCs at various timepoints of IAA treatment (Figure 5C, D). These experiments showed that acetyl-p53 and phospho-p53 are upregulated after 2 hours and 48 hours of auxin treatment, respectively. These data are added to Figure 5C, D and discussed in the Results section.

We also attempted to assess *Chek2* phosphorylation using two different phospho-*Chek2* antibodies (Abnova PAB25281, Cell Signalling #2661). However, we were unable to obtain conclusive results due to non-specificity and non-reactivity of these antibodies in mESCs. However, we clearly show that both, genetic ablation of *Chek2* in *Smc5* cKO mice and inhibition of CHEK2 in SMC-AID depleted mESCs, rescues the phenotypes mediated by SMC5/6 depletion. Thus, we feel that we have provided sufficient evidence that the p53-CHEK2 pathway is a primary response axis following SMC5/6 failure.

Using immunofluorescence analysis, we also assessed the cell cycle distribution of phospho-p53+ mESCs using proliferative cell markers Ki67 and H3P. Ki67 staining demonstrated that close to 100% of cells were proliferative. Based on a prior publication (Mallm and Rippe, 2015) we used H3P expression to define G1 (H3P-), S/G2 and mitotic stages (H3P+) of cell cycle in mESCs and quantified the distribution of phospho-p53 positive cells. Details are provided in the Results section. We observed a global increase in the percentage of phospho-p53+ cells in both H3P+ and H3P- populations after 48 hours of IAA treatment. The abundance of phospho-p53+ cells in prophase and G1 populations compared to S/G2 cells suggests that p53 (S389) phosphorylation is most likely to occur in prophase or G1 phases upon SMC5 depletion. Specifically, 38.2% of G1 phase cells, 26.2% of S/G2 phase cells and 40.8% of prophase cells were positive for p-p53 in the IAA-treated group compared to 11.9%, 11.8% and 7.0% in control, respectively. We also observed a 2.1-fold increase in the percentage of total prophase cells, a 1.5-fold decrease in the percentage of total S/G2 phase cells and a 1.5-fold increase in the percentage of total G1 phase cells in the IAA-treated group compared to control. These data are shown in Figure 5—figure supplement 1D-F and discussed in the Results section.

We also attempted to assess acetyl-p53 immunostaining in mESCs. Unfortunately, we did not see a discernible difference of acetyl-p53 intensity between control and treated mESCs due to a high degree of background staining. Also, we were unable to assess activation of CHEK2 in mESCs due to non-specificity and non-reactivity of the phospho-CHEK2 antibodies we tested (please see above).

Additionally, to validate the reactivity of phospho-CHEK2, phospho-p53 and acetyl-p53 antibodies used for mESC analyses and complement our data, we performed immunofluorescence assessments of mouse E16.5 cortical sections. We found that acetyl-p53 was upregulated in the progenitor zone of *Smc5* cKO cortical sections compared to control. We have added these data to Figure 2—figure supplement 1B and discussed it in the Results section. We did not observe any immunostaining with either the phospho-CHEK2 antibody or phospho-p53 antibody in *Smc5* cKO and control cortical sections due to non-reactivity with this form of immunohistology preparation (with and without antigen retrieval).

In summary, we show that we show the upregulation of acetyl-p53 in *Smc5* cKO cortices and of acetyl-p53 and phospho-p53 in SMC5-depleted mESCs. We demonstrate that acetylation of p53 is an early response to SMC5 depletion in mESCs. In addition, we show that p53 (S389) phosphorylation occurs significantly later than p53 acetylation and is most frequently present in prophase and G1 phase cells. This suggests that aberrancies caused by SMC5 depletion (for example, late replication defects during G2) lead to a DNA damage response during prophase and potentially, the inheritance of DNA damage by G1 daughter cells.

Reviewer #2:This article from the Jordan laboratory describes an extensive set of analyses on a new mouse model where the Smc5 gene was conditionally inactivated. These analyses were complemented by studies on Smc5 degron cells. There have been several previous reports of the effects of impairing Smc5/6 functions in mouse or human cells, but the current study extends those analysis by focusing on effects in vivo and, more specifically, in the developing neocortex. Overall, I really liked the article and the clear manner in which the analyses were described. I think that it will be an important addition to the many times confusing array of studies on Smc5/6 function. There were, however, some concerns with the cellular analyses in particular.1) In several places it is discussed that pericentromeric heterochromatin is a common fragile site. My understanding is that, at least in human cells, this heterochromatic region very rarely, if at all, shows gaps or breaks on mitotic chromosomes. Further, the marker of fragile sites used here, FANCD2, also does not generally localize to pericentromeric regions. I found the discussion around this point quite contrived as if it was intended to link the fact that Smc5/6 localizes to pericentromeric regions with some well recognized manifestation of replication stress like fragility (and MiDAS that they used) when there isn't such a clear association. The authors cite Bhowmick et al. as the reference for EdU foci (MiDAS) and FANCD2 being at centromeric heterochromatin (subsection “SMC5/6 is critical for completion of DNA replication”), but I could find no reference to pericentromeric regions in that paper.

We have edited text according to the reviewer’s comments and included additional references.

Abstract: “Further assessment using *Smc5* cKO and auxin-inducible degron systems demonstrated that absence of SMC5/6 leads to DNA replication stress at late-replicating regions such as pericentromeric heterochromatin regions.”

Results section: “Furthermore, late-replicating regions, such as pericentromeric heterochromatin, are additional sites prone to DNA replication stress (Mendez-Bermudez et al., 2018; Saksouk et al., 2015).”

Results section: “MiDAS often occurs at common fragile sites, AT-rich DNA sequences that are particularly prone to replication stress and subsequent breakage during mitosis, as well as at late-replicating heterochromatin regions (Fungtammasan et al., 2012; Glover et al., 2005; Mendez-Bermudez et al., 2018; Özer and Hickson, 2018; Saksouk et al., 2015).”

2) Related to 1) it would be important to show that real gaps/breaks and MiDAS foci co-localize with at centromeric heterochromatin regions – and that they are distinct from the telomeres that are adjacent in mouse chromosomes. Telomeres are hotspots for fragility and MiDAS. On a related note, the red and pink staining in Figure 6 is really hard to distinguish – couldn't the authors pseudo-color the foci differently?

We have changed FANCD2 foci color into red and centromeres into green in Figure 6B to achieve better color contrast between centromere and EdU signals.

It is important to see where the telomere is on the acrocentric chromosome end. The impression I get is that many of the FANCD2 and EdU foci are at telomeres. These images do not look very crisp either and the resolution could be improved by collecting better confocal images I would suggest.

We include better quality images. We recognize the possibility that on acrocentric chromosomes the foci in proximity to the centromeric signal could be telomeric. We have clarified this in the Results section. In addition, we have quantified the ratio of “acrocentric” EdU foci (that is, foci present on the end of the chromosome containing the centromeres) to “non-acrocentric” EdU foci (foci present on the opposite chromosomal end). This ratio was 1.47 and 1.91 in control and IAA-treated mESCs, respectively. These data have been included in Figure 6—figure supplement 1C and detailed in the Results section. If MiDAS foci occurred solely within the telomeric DNA, they should logically have equal likelihood of occurring at either telomeric end. Thus, the enrichment of acrocentric compared to non-acrocentric foci suggests that features unique to the pericentromeric DNA may confer increased MiDAS susceptibility.

3) Related to 2), what is the percentage of EdU foci in mitosis that co-localize with FANCD2? Unlike what has been shown in previous studies on human cancer cell lines, many of the EdU foci shown here don't seem to co-localize with FANCD2. This should be quantified and discussed in the context of the current literature.

We have quantified the percentage of EdU foci that do not co-localize with FANCD2, which is 59.4% and 56.2% for control and IAA-treated cells, respectively. We also found that 81.3% and 64.5% of FANCD2 foci were EdU- in control and IAA-treated cells, respectively (Figure 6—figure supplement 1B). Thus, FANCD2 foci are less frequently found outside of MiDAS foci upon SMC5 depletion, consistent with the increased number of EdU+FANCD2+ foci seen in IAA-treated cells (Figure 6D). These data have been included in Figure 6—figure supplement 1A, B and discussed in the Results section.

4) The fiber analyses should be repeated with a new control where CHEK2 inhibition is not employed in all of the samples, including the so-called negative control. As things stand, it is hard to compare the current analysis with previous analyses done with controls lacking any added compounds.

We performed additional DNA fiber experiments in the absence or presence of IAA alone (i.e. without CHEK2 inhibitor). We have included these results in Figure 5H and Figure 5—figure supplement 1H, I and discussed it in the Results section. In brief, we show that DNA fibers from IAA-treated mESCs exhibit increased frequency of converging forks and stalled forks, and reduced incidence of ongoing forks, which is similar to the results with CHEK2 inhibition. IAA-treated mESCs also have slightly shorter DNA fiber track length and increased CldU/IdU length ratio, similar to the results with CHEK2 inhibition.